# Last Iterate Convergence of Incremental Methods as a Model of Forgetting

**Xufeng Cai    Jelena Diakonikolas**
Department of Computer Sciences, University of Wisconsin–Madison
xcai74@wisc.edu, jelena@cs.wisc.edu

## Abstract

Incremental gradient and incremental proximal methods are a fundamental class of optimization algorithms used for solving finite sum problems, broadly studied in the literature. Yet, without strong convexity, their convergence guarantees have primarily been established for the ergodic (average) iterate. We establish the first nonasymptotic convergence guarantees for the last iterate of both incremental gradient and incremental proximal methods, in general convex smooth (for both) and convex Lipschitz (for the proximal variants) settings. Our oracle complexity bounds for the last iterate nearly match (i.e., match up to a square-root-log or a log factor) the best known oracle complexity bounds for the average iterate, for both classes of methods. We further obtain generalizations of our results to weighted averaging of the iterates with increasing weights and for randomly permuted ordering of updates. We study last iterate convergence of the incremental proximal method as a mathematical abstraction of forgetting in continual learning and prove a lower bound that certifies that a large amount of regularization is crucial to mitigating catastrophic forgetting—one of the key considerations in continual learning. Our results generalize last iterate guarantees for incremental methods compared to state of the art, as such results were previously known only for overparameterized linear models, which correspond to convex quadratic problems with infinitely many solutions.

## 1 Introduction

We study the last iterate convergence of incremental (gradient and proximal) methods, which apply to problems of the form

$$\min_{\boldsymbol{x} \in \mathbb{R}^d} \Big\{ f(\boldsymbol{x}) := \frac{1}{T} \sum_{t=1}^{T} f_t(\boldsymbol{x}) \Big\}. \tag{1}$$

As is standard, we assume that each component function $f_t$ is convex and either smooth or Lipschitz-continuous and that a minimizer $\boldsymbol{x}_* \in \arg\min_{\boldsymbol{x}} f(\boldsymbol{x})$ exists.

Incremental methods traverse all the component functions $f_t$ in a *cyclic* manner, updating each iterate by taking either a gradient descent step (in the case of incremental gradient methods) or a proximal-point step (in the case of the incremental proximal method) with respect to the individual component functions $f_t$. For a more precise statement of these two classes of methods, see Sections 2 and 3. Same as prior work (Bertsekas et al., 2011; Bertsekas, 2011; Li et al., 2019; Mishchenko et al., 2020; Cai et al., 2024), we define oracle complexity of these methods as the number of first-order or proximal oracle queries to individual component functions $f_t$ required to reach a solution $\boldsymbol{x}$ with optimality gap $f(\boldsymbol{x}) - f(\boldsymbol{x}_*) \leq \epsilon$ on the worst-case instance from the considered problem class, where $\epsilon > 0$ is a given error parameter.

Our main motivation for studying the last iterate convergence of incremental methods comes from its intrinsic abstraction of the catastrophic forgetting in continual learning (CL) with cyclic task replaying. In particular, CL represents a sequential learning setting, where a machine learning model gets updated over time, based on the changing or evolving distribution of the data passed to the learner. A major challenge in such dynamic learning settings is the degradation of model performance on previously seen data, known as the *catastrophic forgetting* (McCloskey & Cohen, 1989;

$$\text{arg min}_x\, f_1(x) + \frac{\|x-x_1\|^2}{2\eta} \qquad \text{arg min}_x\, f_2(x) + \frac{\|x-x_2\|^2}{2\eta} \quad \text{arg min}_x\, f_T(x) + \frac{\|x-x_T\|^2}{2\eta}$$

$$\boxed{\mathsf{x_1}} \longrightarrow \boxed{\mathsf{x_2}} \longrightarrow \cdots \longrightarrow \boxed{\mathsf{x_{T+1}}}$$

Figure 1: Illustration of both continual learning over tasks $\{f_t\}_{t=1}^T$ (cyclic replay + regularization) and incremental proximal method $\boldsymbol{x}_{\cdot,t+1} = \text{prox}_{\eta f_t}(\boldsymbol{x}_{\cdot,t}) = \text{arg min}_{\boldsymbol{x}}\{f_t(\boldsymbol{x}) + \frac{1}{2\eta}\|\boldsymbol{x} - \boldsymbol{x}_{\cdot,t}\|^2\}$.

Goodfellow et al., 2013), which has been well-documented in various empirical studies; see, e.g., recent surveys (Parisi et al., 2019; De Lange et al., 2021). On the theoretical front, however, much is still missing from the understanding of possibilities and limitations related to catastrophic forgetting, with results for basic learning settings being obtained only very recently (Balcan et al., 2015; Evron et al., 2022; Peng & Risteski, 2022; Chen et al., 2022; Cao et al., 2022; Evron et al., 2023; Lin et al., 2023b; Goldfarb & Hand, 2023; Peng et al., 2023; Goldfarb et al., 2024). Our work contributes to this line of theory research, focusing on more general convex settings compared to prior work.

While there are different learning settings studied under the umbrella of CL, following recent work Evron et al. (2022; 2023), we focus on the CL settings with cyclic replaying of tasks. Such settings arise in applications that naturally undergo cyclic changes in the data/tasks, due to diurnal or seasonal cycles (e.g., in agriculture, forestry, e-commerce, astronomy, etc.). For instance, e-commerce platforms like Amazon have predictable seasonal changes in website traffic and purchases around holidays and during promotional periods such as Prime Days.

Given $T$ tasks associated with the loss functions $\{f_t\}_{t=1}^T$, we illustrate the correspondence between the incremental proximal method (IPM) and CL in Fig. 1, where $\boldsymbol{x}_n$ denotes the model parameter in CL or the last iterate of IPM at time $n$. The proximal step of the IPM can be seen as a mathematical abstraction of the training process (minimizing the loss function, with $\ell_2$-regularization enforcing closeness to previous model parameters) on each task. We note in passing that which training algorithm is used to minimize the loss has no bearing on this model, as we are interpreting cyclic replaying of tasks in CL as an optimization algorithm over a discrete set of tasks, similar to previous theoretical work on CL (Evron et al., 2022; 2023; Goldfarb et al., 2024).

The *forgetting* over $T$ tasks after $K$ epochs/cycles in CL with cyclic replaying of tasks is defined by (Doan et al., 2021; Evron et al., 2022; 2023)

$$f_K(\boldsymbol{x}_{K-1,T+1}) := \frac{1}{T}\sum_{t=1}^T f_t(\boldsymbol{x}_{K-1,T+1}), \tag{2}$$

where $\boldsymbol{x}_{k-1,T+1}$ is the model parameter vector after $k^{\text{th}}$ cycle/epoch. Forgetting is *catastrophic* if $f_K(\boldsymbol{x}_{K-1,T+1}) \overset{K\to\infty}{\nrightarrow} 0$.

Observe that Eq. (2) corresponds to the value of the objective function from Eq. (1) at the final iterate $\boldsymbol{x}_K := \boldsymbol{x}_{K-1,T+1}$. Prior work (Evron et al., 2022) that obtained rigorous bounds for the forgetting as in Eq. (2) applied to the problems where each $f_t$ is a convex quadratic function minimized by the same $\boldsymbol{x}_*$ such that $f_t(\boldsymbol{x}_*) = f(\boldsymbol{x}_*) = 0$. By contrast, we consider more general convex functions that are either smooth or Lipschitz continuous, and make no assumption about $\boldsymbol{x}_*$ beyond being a minimizer of the (average) function $f$. Since we are not assuming that $f(\boldsymbol{x}_*) = 0$, our focus is on bounding the excess forgetting $f_K(\boldsymbol{x}_K) - f(\boldsymbol{x}_*)$, equivalently the optimality gap for the last iterate in Eq. (1).

The method considered in Evron et al. (2022) minimized each component function exactly, outputting the solution closest to the previous iterate in each iteration, using implicit regularization properties of SGD. To obtain the results, it was then crucial that the component functions were quadratic (so that there is an explicit, closed-form solution for each subproblem) and that all component functions shared a nonempty set of minimizers with value zero (so that forgetting can be controlled despite aggressive adaption to the current task). Our work using the IPM instead considers *explicit* regularization to enforce closeness of models on differing tasks. This could potentially degrade the performance on the current task, but as a trade-off can control forgetting and it addresses a much broader class of loss functions, without any shared minima assumptions. Such explicit regularization is also motivated by the existing CL practice (Heckel, 2022; Li et al., 2023), as one of major empirical approaches to mitigating catastrophic forgetting (De Lange et al., 2021).

Due to space constraints, further discussion of related work is provided in Appendix A.

## 1.1 CONTRIBUTIONS

Our contributions are twofold. First, we derive the first results for the last iterate convergence rate of standard incremental methods[1]. Second, our results for convergence of the IPM have a direct implication on the catastrophic forgetting in CL, and we formally characterize the effect of regularization on forgetting. Below, we summarize our main contributions, where $\sigma_*^2 := \frac{1}{T}\sum_{t=1}^{T}\|\nabla f_t(\boldsymbol{x}_*)\|^2$ denotes the gradient variance at $\boldsymbol{x}_*$. The quantity $\sigma_*^2$ is intrinsic to oracle complexity of incremental methods (Mishchenko et al., 2020; Nguyen et al., 2021; Cai et al., 2024; Cha et al., 2023).

**Last iterate convergence of Incremental Gradient Descent (IGD).** We provide the first oracle complexity guarantees for the last iterate of standard variants of IGD with either deterministic or randomly permuted ordering of the updates, applied to convex $L$-smooth objectives. Up to a square-root-log factor, our oracle complexity bounds in Theorem 1 and Corollary 2— which are $\widetilde{\mathcal{O}}\big(\frac{TL\|\boldsymbol{x}_0-\boldsymbol{x}_*\|^2}{\epsilon} + \frac{TL^{1/2}\sigma_*\|\boldsymbol{x}_0-\boldsymbol{x}_*\|^2}{\epsilon^{3/2}}\big)$ for the deterministic variant and $\widetilde{\mathcal{O}}\big(\frac{TL\|\boldsymbol{x}_0-\boldsymbol{x}_*\|^2}{\epsilon} + \frac{\sqrt{TL}\sigma_*\|\boldsymbol{x}_0-\boldsymbol{x}_*\|^2}{\epsilon^{3/2}}\big)$ for the randomly permuted variant—match the best known oracle complexity bounds for these methods, previously known only for the (uniformly) average iterate (Mishchenko et al., 2020; Nguyen et al., 2021; Cai et al., 2024; Cha et al., 2023). We further extend our results to increasing weighted averaging of the iterates in Corollary 1, which places more weight on the more recent iterates, removing the excess square-root-log factor in the resulting oracle complexity bound.

**Last iterate convergence of Incremental Proximal Method (IPM).** We provide the first oracle complexity guarantees for the last iterate of IPM applied to convex and either smooth or Lipschitz-continuous objectives. When each component function is convex and $L$-smooth, we show (in Theorem 2) that IPM has the same $\widetilde{\mathcal{O}}\big(\frac{TL\|\boldsymbol{x}_0-\boldsymbol{x}_*\|^2}{\epsilon} + \frac{TL^{1/2}\sigma_*\|\boldsymbol{x}_0-\boldsymbol{x}_*\|^2}{\epsilon^{3/2}}\big)$ oracle complexity as IGD. This result is new for any variant of this method—with average or last iterate as its output. When component functions are convex and $G$-Lipschitz, our oracle complexity $\widetilde{\mathcal{O}}\big(\frac{G^2T\|\boldsymbol{x}_0-\boldsymbol{x}_*\|^2}{\epsilon^2}\big)$ in Theorem 4 matches the best known oracle complexity bound up to a log factor, which was previously known only for the (uniformly) average iterate (Bertsekas, 2011; Li et al., 2019). We further argue (in Corollary 3 and Corollary 4) that for both settings our analysis can be extended to admit *inexact* proximal point evaluations—an important setting not addressed by prior work on general IPM.

**IPM as a model of CL.** We initiate the study of IPM as a model of CL, corresponding to sequential ridge-regularized model training commonly used in practice. On the positive side, our last-iterate convergence results for IPM in Theorem 2 and Theorem 3 demonstrate that forgetting (corresponding to the optimality gap at the last iterate) can be effectively controlled if the amount of employed regularization is sufficiently high. On the negative side, we show that for any constant amount of regularization, forgetting is always catastrophic, even for least squares problems. In particular, we provide a univariate quadratic example such that for any constant regularization parameter, the asymptotic limit of (excess) forgetting is non-zero. Further, we prove that for forgetting to be made smaller than some target $\epsilon$, the regularization must be sufficiently high and depend polynomially on $1/\epsilon$, $T$, and $\sigma_*$. These results are summarized in Theorem 3 and highlight the limitations of regularization as a black-box tool for controlling forgetting in CL.

## 1.2 PRELIMINARIES

We consider the $d$-dimensional real vector space $(\mathbb{R}^d, \|\cdot\|)$, where $\|\cdot\|$ is the $\ell_2$ norm. We denote $[T] := \{1, 2, \dots, T\}$. Given a proper, convex, lower semicontinuous function $f$, its proximal operator and Moreau envelope are defined by

$$\text{prox}_{\eta f}(\boldsymbol{x}) = \arg\min_{\boldsymbol{y}\in\mathbb{R}^d}\big\{\tfrac{1}{2\eta}\|\boldsymbol{y}-\boldsymbol{x}\|^2 + f(\boldsymbol{y})\big\}, \ M_{\eta f}(\boldsymbol{x}) = \min_{\boldsymbol{y}\in\mathbb{R}^d}\big\{\tfrac{1}{2\eta}\|\boldsymbol{y}-\boldsymbol{x}\|^2 + f(\boldsymbol{y})\big\},$$

---

[1]An independent and concurrent work to ours (Liu & Zhou, 2024b) studied the last iterate convergence of shuffled SGD for composite (strongly) convex smooth/Lipschitz optimization. For the same problems as studied in our Section 2, they obtained the same convergence results. The remaining results in (Liu & Zhou, 2024b) and our work are not directly comparable, as the motivation for the two papers and the studied settings are different; for a more detailed discussion, see Liu & Zhou (2024b, Section 2) and our Appendix A.

---

**Algorithm 1** Incremental Gradient Descent (IGD)

---

    **Input:** initial point $\boldsymbol{x}_0$, number of epochs $K$, step size $\{\eta_k\}$
    **for** $k = 1 : K$ **do**
        $\boldsymbol{x}_{k-1,1} = \boldsymbol{x}_{k-1}$
        **for** $t = 1 : T$ **do**
            $\boldsymbol{x}_{k-1,t+1} = \boldsymbol{x}_{k-1,t} - \eta_k \nabla f_t(\boldsymbol{x}_{k-1,t})$
        $\boldsymbol{x}_k = \boldsymbol{x}_{k-1,T+1}$
    **return** $\boldsymbol{x}_K$

---

respectively, for a parameter $\eta > 0$. The Moreau envelope is $\frac{1}{\eta}$-smooth with the gradient $\nabla M_{\eta f}(\boldsymbol{x}) = \frac{1}{\eta}(\boldsymbol{x} - \mathrm{prox}_{\eta f}(\boldsymbol{x})) \in \partial f(\mathrm{prox}_{\eta f}(\boldsymbol{x}))$, where $\partial f(\cdot)$ is the subdifferential of $f$.

We make the following assumptions. The first one is made throughout the paper.

**Assumption 1.** *Each $f_t$ is convex and there exists a minimizer $\boldsymbol{x}_* \in \arg\min_{\boldsymbol{x} \in \mathbb{R}^d} f(\boldsymbol{x})$.*

By Assumption 1, $f$ is also convex. In nonsmooth settings, we make an additional standard assumption that the component functions are Lipschitz-continuous.

**Assumption 2.** *Each $f_t$ is $G$-Lipschitz, i.e., $|f_t(\boldsymbol{x}) - f_t(\boldsymbol{y})| \leq G\|\boldsymbol{x} - \boldsymbol{y}\|$ for any $\boldsymbol{x}, \boldsymbol{y} \in \mathbb{R}^d$; thus $\|g_t(\boldsymbol{x})\| \leq G$ for all $g_t(\boldsymbol{x}) \in \partial f_t(\boldsymbol{x})$.*

For the smooth settings, we make the following assumption.

**Assumption 3.** *Each $f_t$ is $L$-smooth, i.e., $\|\nabla f_t(\boldsymbol{x}) - \nabla f_t(\boldsymbol{y})\| \leq L\|\boldsymbol{x} - \boldsymbol{y}\|$ for any $\boldsymbol{x}, \boldsymbol{y} \in \mathbb{R}^d$.*

We remark that Assumptions 2 and 3 imply that $f$ is also $G$-Lipschitz and $L$-smooth, respectively. These two assumptions can also be generalized to be with distinct Lipschitz/smoothness constants, and our results would scale with the average Lipschitz/smoothness constant using the techniques from Cai et al. (2024), which we omit to keep the focus on the intricacies of the last iterate convergence. When $f$ is $L$-smooth and convex, we will often make use of the following standard inequality that fully characterizes the class of $L$-smooth convex functions:

$$\frac{1}{2L}\|\nabla f(\boldsymbol{x}) - \nabla f(\boldsymbol{y})\|^2 \leq f(\boldsymbol{y}) - f(\boldsymbol{x}) - \langle \nabla f(\boldsymbol{x}), \boldsymbol{y} - \boldsymbol{x} \rangle, \quad \forall \boldsymbol{x}, \boldsymbol{y} \in \mathbb{R}^d. \tag{3}$$

Finally, when each $f_t$ is smooth, we assume bounded variance at $\boldsymbol{x}_*$, same as all prior work that considered the same settings of IGD/shuffled SGD as we do (Mishchenko et al., 2020; Nguyen et al., 2021; Tran et al., 2021; 2022; Cai et al., 2024).

**Assumption 4.** *The quantity $\sigma_*^2 := \frac{1}{T}\sum_{t=1}^{T} \|\nabla f_t(\boldsymbol{x}_*)\|^2$ is bounded.*

## 2   LAST ITERATE CONVERGENCE OF INCREMENTAL GRADIENT DESCENT

In this section, we introduce our techniques for analyzing the last iterate guarantee and bound the oracle complexity for the last iterate of incremental gradient descent (IGD), assuming component functions are smooth and convex. In the context of CL, this corresponds to a simplified setup where the learner incrementally performs a single gradient step on each task and cyclically replays the $T$ tasks. Nevertheless, this setup serves as a warmup to the proximal setup we discuss in the next section. Additionally, it is of independent interest as incremental gradient methods are widely used in the optimization and machine learning literature, where despite the lack of prior theoretical justification, it is typically the last iterate that gets output by the algorithm in practice.

We summarize the IGD method in Alg. 1, assuming the incremental order $1, 2, \ldots, T$ in each epoch for simplicity and without loss of generality. The oracle complexity for the (uniformly) average iterate of IGD has been shown to be $\mathcal{O}(\frac{TL}{\epsilon} + \frac{T\sqrt{L}\sigma_*}{\epsilon^{3/2}})$ for an $\epsilon$-optimality gap (Mishchenko et al., 2020; Cai et al., 2024) under the same assumptions we make here (Assumptions 1, 3, and 4), while, as discussed before, there were no guarantees for either the last iterate or even a weighted average of the iterates. The main result of this section is that the same oracle complexity applies to the last iterate of IGD, up to a square-root-log factor. We further generalize this result to weighted averages of iterates with increasing weights and to variants with randomly permuted order of cyclic updates.

We begin the analysis by deriving a bound on the gap with respect to an arbitrary but fixed reference point $\boldsymbol{z}$, as summarized in the following lemma, whose proof is in Appendix B. This stands in contrast to arguments deriving bounds on the average iterate, which take $\boldsymbol{z} = \boldsymbol{x}_*$. While this may seem like a minor difference, it affects the analysis non-trivially: a direct extension of prior arguments would require replacing Assumption 4—which imposes a bound on $\frac{1}{T}\sum_{t=1}^T \|\nabla f_t(\boldsymbol{x}_*)\|^2$—with a bound on $\frac{1}{T}\sum_{t=1}^T \|\nabla f_t(\boldsymbol{z})\|^2$ for an arbitrary $\boldsymbol{z}$, which would be a much stronger requirement.

**Lemma 1.** *Under Assumptions 1 and 3, for any $\boldsymbol{z} \in \mathbb{R}^d$ that is fixed in the $k$-th cycle of Alg. 1 and any $\alpha, \beta > 0$ such that $\frac{1}{\alpha} + \frac{1}{\beta} \leq \frac{1}{2}$, if $\eta_k \leq \frac{1}{\sqrt{\beta}TL}$, then for all $k \in [K]$,*

$$T\big(f(\boldsymbol{x}_k) - f(\boldsymbol{z})\big) \leq \eta_k^2 L \sum_{t=1}^T \big\| \sum_{s=t}^T \nabla f_s(\boldsymbol{x}_*) \big\|^2 + \frac{\alpha}{\beta}T\big(f(\boldsymbol{z}) - f(\boldsymbol{x}_*)\big)$$
$$+ \frac{1}{2\eta_k}\big(\|\boldsymbol{x}_{k-1} - \boldsymbol{z}\|^2 - \|\boldsymbol{x}_k - \boldsymbol{z}\|^2\big).$$

Our next step is to specify our choice of the reference point $\boldsymbol{z}$ for each epoch. In particular, we consider a sequence of points $\{\boldsymbol{z}_k\}_{k=-1}^{K-1}$ that is recursively defined as a convex combination of the algorithm iterate $\boldsymbol{x}_k$ and the previous reference point $\boldsymbol{z}_{k-1}$:

$$\boldsymbol{z}_k = (1 - \lambda_k)\boldsymbol{x}_k + \lambda_k \boldsymbol{z}_{k-1} \tag{4}$$

for $k \geq 0$ with $\boldsymbol{z}_{-1} = \boldsymbol{x}_*$ and $\lambda_k \in [0, 1]$ to be set later. Observe that $\boldsymbol{z}_k$ can also be written as a convex combination of the points $\{\boldsymbol{x}_j\}_{j=0}^k$ and $\boldsymbol{x}_*$ by unrolling the recursion, i.e.,

$$\boldsymbol{z}_k = (1 - \lambda_k)\boldsymbol{x}_k + \big(\textstyle\prod_{i=0}^k \lambda_i\big)\boldsymbol{x}_* + \sum_{j=0}^{k-1}\big(\prod_{i=j+1}^k \lambda_i\big)(1 - \lambda_j)\boldsymbol{x}_j, \tag{5}$$

where $(1 - \lambda_k) + \prod_{i=0}^k \lambda_i + \sum_{j=0}^{k-1}\big(\prod_{i=j+1}^k \lambda_i\big)(1 - \lambda_j) = 1$. If we set $\lambda_k = 1$ for all $k$, then we have $\boldsymbol{z}_k = \boldsymbol{x}_*$ and recover the bound $f(\boldsymbol{x}_k) - f(\boldsymbol{x}_*)$ in Lemma 1, which leads to the average iterate guarantee. For general $\{\lambda_k\}$, we obtain the following lemma to relate the function value gap $f(\boldsymbol{x}_k) - f(\boldsymbol{z}_{k-1})$ to the optimality gap $f(\boldsymbol{x}_k) - f(\boldsymbol{x}_*)$, whose proof is deferred to Appendix B.

**Lemma 2.** *Let $\boldsymbol{z}_k$ be defined by Eq. (4) for a given sequence of parameters $\lambda_k \in (0, 1)$, where $k \geq 0$ and $\boldsymbol{z}_{-1} = \boldsymbol{x}_*$. Under Assumption 1, if there exists a sequence of nonnegative weights $w_k$ such that $\lambda_k w_k \leq w_{k-1}$ for $k \in [K - 1]$, then for all $k \in [K]$:*

*1.* $w_{k-1}\big(f(\boldsymbol{z}_{k-1}) - f(\boldsymbol{x}_*)\big) \leq \sum_{j=0}^{k-1} w_j(1 - \lambda_j)\big(f(\boldsymbol{x}_j) - f(\boldsymbol{x}_*)\big);$

*2.* $w_{k-1}\big(f(\boldsymbol{x}_k) - f(\boldsymbol{z}_{k-1})\big) \geq w_{k-1}(f(\boldsymbol{x}_k) - f(\boldsymbol{x}_*)) - \sum_{j=0}^{k-1} w_j(1 - \lambda_j)(f(\boldsymbol{x}_j) - f(\boldsymbol{x}_*)).$

The role of the sequence of weights $\{w_k\}$ in Lemma 2 is to ensure that we can telescope the terms $\|\boldsymbol{x}_{k-1} - \boldsymbol{z}_{k-1}\|^2 - \|\boldsymbol{x}_k - \boldsymbol{z}_{k-1}\|^2$ in Lemma 1, when summing the weighted per-epoch recursion $w_{k-1}\big(f(\boldsymbol{x}_k) - f(\boldsymbol{z}_{k-1})\big)$ over $k \in [K]$ and deriving the convergence results. On the other hand, to succinctly see why such $\{\boldsymbol{z}_k\}$ could lead to the desired last iterate guarantees, we note that the second part of Lemma 2 indicates that $w_{k-1}\big(f(\boldsymbol{x}_k) - f(\boldsymbol{z}_{k-1})\big)$ intrinsically includes retraction terms of the optimality gaps at the previous iterates. Thus, we can deduct $w_{K-1}(f(\boldsymbol{x}_K) - f(\boldsymbol{x}_*))$ from $\sum_{k=1}^K w_{k-1}\big(f(\boldsymbol{x}_k) - f(\boldsymbol{z}_{k-1})\big)$ by properly choosing $\lambda_k$ and $w_k$ to cancel out the optimality gap terms at the intermediate iterates. In this case, the convergence rate for the last iterate would be characterized by $\mathcal{O}\big(\sum_{k=1}^K w_{k-1}/w_{K-1}\big)$.

Our choice of $\{\boldsymbol{z}_k\}$ is inspired by the recent work (Zamani & Glineur, 2023; Liu & Zhou, 2024a) on last iterate guarantees for subgradient methods and SGD with $\lambda_k = \frac{w_{k-1}}{w_k}$. However, their proof techniques are not directly applicable to incremental methods, due to technical obstacles including additional nontrivial error terms of the form $\frac{\alpha}{\beta}T\big(f(\boldsymbol{z}_k) - f(\boldsymbol{x}_*)\big)$ in Lemma 1 arising from the incremental gradient steps using reference points other than $\boldsymbol{x}_*$. Further discussion is in Appendix A. Such error terms inherently deteriorate the growth rate of $\{w_k\}$ and could lead to a worse last iterate rate compared to the rate on the average iterate. In the following theorem, we calibrate such degradation on the last iterate guarantees, and show that with a slightly smaller step size one can still achieve essentially the same rate as for the average iterate. The proof is provided in Appendix B.

**Theorem 1.** *Under Assumptions 1, 3, and 4 and for positive parameters $\alpha, \beta > 0$ such that $\frac{1}{\alpha} + \frac{1}{\beta} \leq \frac{1}{2}$, if the step size is fixed and satisfies $\eta_k = \eta \leq \frac{1}{\sqrt{\beta}TL}$, the output $\boldsymbol{x}_K$ of Alg. 1 satisfies*

$$f(\boldsymbol{x}_K) - f(\boldsymbol{x}_*) \leq \mathrm{e}\eta^2 T^2 \sigma_*^2 L(1 + \beta/\alpha)K^{\frac{\alpha/\beta}{1+\alpha/\beta}} + \frac{\mathrm{e}\|\boldsymbol{x}_0 - \boldsymbol{x}_*\|^2}{2\eta T K^{\frac{1}{1+\alpha/\beta}}}. \tag{6}$$

*In particular, for $\alpha = 4, \beta = 4\log K$, and $\eta = \min\left\{\frac{\|\boldsymbol{x}_0 - \boldsymbol{x}_*\|^{2/3}}{2^{1/3}T\sigma_*^{2/3}L^{1/3}K^{1/3}(1+\log K)^{1/3}}, \frac{1}{2\sqrt{\log K}TL}\right\}$,*

$f(\boldsymbol{x}_K) - f(\boldsymbol{x}_*) = \mathcal{O}\left(\frac{L^{1/3}\sigma_*^{2/3}\|\boldsymbol{x}_0 - \boldsymbol{x}_*\|^{4/3}(1+\log K)^{1/3}}{K^{2/3}} + \frac{L\|\boldsymbol{x}_0 - \boldsymbol{x}_*\|^2\sqrt{\log K}}{K}\right)$, *and thus* $f(\boldsymbol{x}_K) -$ $f(\boldsymbol{x}_*) \leq \epsilon$ *after* $\widetilde{\mathcal{O}}\left(\frac{TL\|\boldsymbol{x}_0 - \boldsymbol{x}_*\|^2}{\epsilon} + \frac{TL^{1/2}\sigma_*\|\boldsymbol{x}_0 - \boldsymbol{x}_*\|^2}{\epsilon^{3/2}}\right)$ *individual gradient evaluations.*

The error term $f(\boldsymbol{z}_{k-1}) - f(\boldsymbol{x}_*)$ in Lemma 1 plays the role of slowing the last iterate rate, as calibrated by the dependence on $\alpha/\beta$ in Eq. (6). To remedy such degradation compared to the average iterate rate (Mishchenko et al., 2020; Cai et al., 2024), one natural thought is to make $\alpha/\beta$ sufficiently small. In particular, we choose $\alpha/\beta = 1/\log K$ and show that the last iterate rate nearly matches the best known rate on the average iterate, with the trade-off of requiring order-$\frac{1}{\sqrt{\log K}}$ smaller step sizes in comparison with Mishchenko et al. (2020); Cai et al. (2024). This translates into the oracle complexity that is larger by at most a $\sqrt{\log(1/\epsilon)}$ factor. For most cases of interest, this quantity can be treated as a constant: for example, for $\epsilon = 10^{-8}$, $\sqrt{\log(1/\epsilon)} \approx 4.29$.

On the other hand, with $\lambda_k \equiv 1$ and constant $w_k$, Lemmas 1 and 2 directly imply the average iterate guarantee, as a sanity check. Additionally, instead of zeroing the weights of the optimality gap terms $f(\boldsymbol{x}_k) - f(\boldsymbol{x}_*)$ for $k \in [K-1]$ to obtain the last iterate guarantee, one can deduce the convergence rate on the increasing weighted averaging that places more weight on later iterates, as formalized in the following corollary. The proof is deferred to Appendix B.

**Corollary 1** (Increasing Weighted Averaging). *Under Assumptions 1, 3, and 4 and for parameters $\alpha, \beta > 0$ such that $\frac{1}{\alpha} + \frac{1}{\beta} \leq \frac{1}{2}$, if the step size $\eta$ is fixed and such that $\eta \leq \frac{1}{\sqrt{\beta}TL}$, then for any constant $c \in (0, 1]$ and increasing sequence $\{w_k\}_{k=0}^{K-1}$ with $w_k = \frac{(1+\alpha/\beta)(K-k)+1-c}{(1+\alpha/\beta)(K-k)}w_{k-1}$, Alg. 1 outputs $\hat{\boldsymbol{x}}_K = \sum_{k=1}^{K}\frac{w_{k-1}\boldsymbol{x}_k}{\sum_{k=1}^{K}w_{k-1}}$ satisfying*

$$f(\hat{\boldsymbol{x}}_K) - f(\boldsymbol{x}_*) \leq \frac{T^2\sigma_*^2\eta^2 L}{c} + \frac{\|\boldsymbol{x}_0 - \boldsymbol{x}_*\|^2}{2c\eta TK}.$$

*In particular, taking $\beta = \mathcal{O}(1)$ and the step size $\eta = \min\left\{\frac{1}{\sqrt{\beta}TL}, \left(\frac{\|\boldsymbol{x}_0 - \boldsymbol{x}_*\|^2}{2T^3\sigma_*^2LK}\right)^{1/3}\right\}$, we have $f(\hat{\boldsymbol{x}}_K) - f(\boldsymbol{x}_*) \leq \epsilon$ after $\mathcal{O}\left(\frac{TL\|\boldsymbol{x}_0 - \boldsymbol{x}_*\|^2}{c\epsilon} + \frac{TL^{1/2}\sigma_*\|\boldsymbol{x}_0 - \boldsymbol{x}_*\|^2}{c^{3/2}\epsilon^{3/2}}\right)$ individual gradient evaluations.*

We remark that increasing weighted averaging shaves off the (at most) square-root-log term appearing in the last iterate rate above and recovers the best known rate for the average iterate (Mishchenko et al., 2020; Cai et al., 2024). The parameters $\alpha, \beta$ are included in the weights $w_k$ controlling the growth rate of the increasing sequence $\{w_k\}$. When $\alpha/\beta \to \infty$, increasing weighted averaging reduces to the uniform weighted average.

**Shuffled SGD.** We extend our analysis to handle the case with possible random permutations on the task ordering in each epoch, showing order-$\sqrt{T}$ improvements in complexity if involving randomness. We consider two main permutation strategies of particular interests in the literature on shuffled SGD: (i) random reshuffling (RR): randomly generate permutations at the beginning of each epoch; (ii) shuffle-once (SO): generate a single random permutation at the beginning and use it in all epochs. Those strategies lead to order-$(1/T)$ improvements in bounding the variance term $\eta_k^2 L \sum_{t=1}^{T}\left\|\sum_{s=t}^{T}\nabla f_s(\boldsymbol{x}_*)\right\|^2$ from Lemma 1, and we state the improved convergence results with permutations in the following corollary. The proof is deferred to Appendix B. Our last iterate guarantee nearly matches the best known average iterate convergence rate for shuffled SGD (Mishchenko et al., 2020; Nguyen et al., 2021; Cai et al., 2024) and the lower bound results for the RR scheme (Cha et al., 2023), with a slightly (order-$(\frac{1}{\sqrt{\log K}})$) smaller step size.

**Corollary 2** (Shuffled SGD (RR/SO)). *Under Assumptions 1, 3 and 4 and for positive parameters $\alpha, \beta > 0$ such that $\frac{1}{\alpha} + \frac{1}{\beta} \leq \frac{1}{2}$, if the step size is fixed and such that $\eta \leq \frac{1}{\sqrt{\beta}TL}$, the output $\boldsymbol{x}_K$ of Alg. 1 with uniformly random (SO/RR) shuffling satisfies*

$$\mathbb{E}[f(\boldsymbol{x}_K) - f(\boldsymbol{x}_*)] \leq e\eta^2\sigma_*^2 TL(1 + \beta/\alpha)K^{\frac{\alpha/\beta}{1+\alpha/\beta}} + \frac{e\|\boldsymbol{x}_0 - \boldsymbol{x}_*\|^2}{2T\eta K^{\frac{1}{1+\alpha/\beta}}}.$$

*With $\alpha = 4, \beta = 4\log K$ and $\eta = \min\left\{\frac{\|\boldsymbol{x}_0 - \boldsymbol{x}_*\|^{2/3}}{2^{1/3}T^{2/3}\sigma_*^{2/3}L^{1/3}K^{1/3}(1+\log K)^{1/3}}, \frac{1}{2\sqrt{\log K}TL}\right\}$, we have*

$$\mathbb{E}[f(\boldsymbol{x}_K) - f(\boldsymbol{x}_*)] \leq \epsilon \text{ after } \widetilde{\mathcal{O}}\left(\frac{TL\|\boldsymbol{x}_0 - \boldsymbol{x}_*\|^2}{\epsilon} + \frac{\sqrt{TL}\sigma_*\|\boldsymbol{x}_0 - \boldsymbol{x}_*\|^2}{\epsilon^{3/2}}\right) \text{ individual gradient evaluations.}$$

---

**Algorithm 2** Incremental Proximal Method (IPM)

**Input:** initial point $\boldsymbol{x}_0$, number of epochs $K$, step size $\{\eta_k\}$
**for** $k = 1 : K$ **do**
    $\boldsymbol{x}_{k-1,1} = \boldsymbol{x}_{k-1}$
    **for** $t = 1 : T$ **do**
        $\boldsymbol{x}_{k-1,t+1} = \text{prox}_{\eta_k f_t}(\boldsymbol{x}_{k-1,t}) = \arg\min_{\boldsymbol{x} \in \mathbb{R}^d} \left\{ \frac{1}{2\eta_k} \|\boldsymbol{x} - \boldsymbol{x}_{k-1,t}\|^2 + f_t(\boldsymbol{x}) \right\}$
    $\boldsymbol{x}_k = \boldsymbol{x}_{k-1,T+1}$
**return** $\boldsymbol{x}_K$

---

It is worth noting here that the analysis of shuffled SGD is technically disjoint from the traditional analysis of SGD, which relies on sampling with replacement. (By contrast, shuffled SGD employs sampling *without* replacement.) As a consequence, the last iterate results for SGD are not applicable to incremental methods (nor shuffled SGD), as discussed in more details in Appendix A.

## 3 INCREMENTAL PROXIMAL METHOD

In this section, we leverage the proof techniques developed in the previous section and derive the last iterate convergence bounds for the IPM, summarized in Alg. 2. While the IPM is a fundamental method broadly studied in the optimization literature—and thus its last iterate convergence bounds are of independent interest—our main motivation for considering this method comes from using it as a mathematical abstraction of CL, as discussed in the introduction. Thus we begin this section by briefly explaining this connection and reasoning.

The considered setup is motivated by the general $\ell_2$-regularized CL setting (Heckel, 2022; Li et al., 2023). In particular, each proximal iteration $\boldsymbol{x}_{k-1,t+1} = \text{prox}_{\eta_k f_t}(\boldsymbol{x}_{k-1,t})$ can be interpreted as minimizing the $\ell_2$ (a.k.a. ridge) regularized loss $f_t(\boldsymbol{x}) + \frac{1}{2\eta_k}\|\boldsymbol{x} - \boldsymbol{x}_{k-1,t}\|_2^2$ corresponding to the current task $t$, which aligns with the common machine learning practice of using regularization to improve the generalization error and prevent forgetting. When $\eta_k \to \infty$, the proximal point step reduces to the CL setting where the learner exactly minimizes the loss of the current task, i.e., $\boldsymbol{x}_{k-1,t+1} = \arg\min_{\boldsymbol{x} \in \mathbb{R}^d} f_t(\boldsymbol{x})$, while the regularization effect vanishes and causes larger forgetting on previous tasks. When $\eta_k$ is small, the proximal point step is easier to compute with larger quadratic regularization and prevents deviating from the previous iterate thus causing less forgetting. However, in this case the plasticity of the model may be deteriorated.

We also note that our analysis of IPM is related to previous work on cyclic replays for overparameterized linear models with (S)GD (Evron et al., 2022; Swartworth et al., 2024), as (S)GD in this case acts as an implicit regularizer (Gunasekar et al., 2018; Zhang et al., 2021) (whereas proximal point update acts as an *explicit* regularizer). The two lines of work are not directly comparable: Evron et al. (2022); Swartworth et al. (2024) considers exact minimization of component/task loss function and bounds the forgetting, but only addresses convex quadratics where the component loss functions have a nonempty intersecting set of minima (which implies $\sigma_* = 0$ in Assumption 4). On the other hand, our work addresses much more general (not necessarily quadratic) convex functions and does not require $\sigma_* = 0$, but instead relies on sufficiently large regularization.

### 3.1 SMOOTH CONVEX SETTING

We first study the setting where the loss of each task is convex and smooth, for which we show faster convergence and which covers many regression tasks studied in prior CL work; see e.g., Evron et al. (2022); Goldfarb et al. (2024). In contrast, prior work either only focused on nonsmooth settings for IPM (Bertsekas, 2011; Li et al., 2019; 2020) or studied different algorithms without component-wise proximal steps for smooth settings (Bertsekas, 2015; Mishchenko et al., 2022).

Under component smoothness, the proximal iteration is equivalent to the backward gradient step:

$$\boldsymbol{x}_{k-1,t+1} = \boldsymbol{x}_{k-1,t} - \eta_k \nabla f_t(\boldsymbol{x}_{k-1,t+1}).$$

Hence, much of the analysis from Section 2 can be adapted here, and the main difference lies in bounding the gap $f(\boldsymbol{x}_k) - f(\boldsymbol{z})$ within each epoch with decomposition w.r.t. $\boldsymbol{x}_{k-1,t+1}$ instead of

$\boldsymbol{x}_{k-1,t}$ in comparison to Lemma 1. Then choosing $\boldsymbol{z} = \boldsymbol{z}_{k-1}$ defined by Eq. (4) and mimicking the proof of Theorem 1, we obtain the following theorem, whose proof is in Appendix C.

**Theorem 2.** *Under Assumptions 1, 3, and 4 and for parameters $\alpha, \beta > 0$ such that $\frac{1}{\alpha} + \frac{1}{\beta} \leq \frac{1}{2}$, if the step size is fixed and such that $\eta \leq \frac{1}{\sqrt{\beta}TL}$, the output $\boldsymbol{x}_K$ of Alg. 2 satisfies*

$$f(\boldsymbol{x}_K) - f(\boldsymbol{x}_*) \leq \mathrm{e}\eta^2 T^2 \sigma_*^2 L(1 + \beta/\alpha) K^{\frac{\alpha/\beta}{1+\alpha/\beta}} + \frac{\mathrm{e}\|\boldsymbol{x}_0 - \boldsymbol{x}_*\|^2}{2\eta T K^{\frac{1}{1+\alpha/\beta}}}.$$

*With $\alpha = 4, \beta = 4\log K$ and $\eta = \min\left\{\frac{\|\boldsymbol{x}_0 - \boldsymbol{x}_*\|^{2/3}}{2^{1/3}T\sigma_*^{2/3}L^{1/3}K^{1/3}(1+\log K)^{1/3}}, \frac{1}{2\sqrt{\log K}TL}\right\}$, we have*

$f(\boldsymbol{x}_K) - f(\boldsymbol{x}_*) = \mathcal{O}\left(\frac{L^{1/3}\sigma_*^{2/3}\|\boldsymbol{x}_0 - \boldsymbol{x}_*\|^{4/3}(1+\log K)^{1/3}}{K^{2/3}} + \frac{L\|\boldsymbol{x}_0 - \boldsymbol{x}_*\|^2\sqrt{\log K}}{K}\right)$, *and thus* $f(\boldsymbol{x}_K) - f(\boldsymbol{x}_*) \leq \epsilon$ *after* $\widetilde{\mathcal{O}}\left(\frac{TL\|\boldsymbol{x}_0 - \boldsymbol{x}_*\|^2}{\epsilon} + \frac{TL^{1/2}\sigma_*\|\boldsymbol{x}_0 - \boldsymbol{x}_*\|^2}{\epsilon^{3/2}}\right)$ *individual gradient evaluations.*

A few remarks are in order here. First, the last iterate convergence rate of IPM matches the rate of IGD for the last iterate. This is also the first convergence result for IPM (with component-wise proximal updates) in smooth convex settings, in comparison with the prior results for convex Lipschitz setups (Bertsekas, 2011; Li et al., 2019; 2020). Second, the extensions to the increasing weighted averaging and RR/SO shuffling discussed in Section 2 also apply to this setting, which we omit for brevity. Lastly, the step size constraint in Theorem 2 may seem surprising, given that the proximal point method (corresponding to IPM with $T = 1$) converges for any positive step size. However, as we show in the following theorem, such a step size restriction is necessary for IPM (with $T \geq 2$) to reach the target optimality gap, highlighting challenges arising from incremental updates. We provide a proof sketch below, while the complete proof is in Appendix C.

**Theorem 3.** *Given $L > 0$ and $T \geq 2$, let $\mathcal{F}_{T,L}$ be the class of finite-sum functions $f(\boldsymbol{x}) = \frac{1}{T}\sum_{t=1}^T f_t(\boldsymbol{x})$, with each component $f_t$ being $L$-smooth and convex. Then:*

*1. For any fixed step size $\eta$, there exists a function $f \in \mathcal{F}_{T,L}$ such that the iterates $\boldsymbol{x}_k$ of Alg. 2 satisfy $f(\boldsymbol{x}_k) - f(\boldsymbol{x}_*) \not\to 0$ as $k \to \infty$. As a consequence, the forgetting is catastrophic.*

*2. For any fixed step size $\eta$ that only depends on the parameters of the problem class ($L, T$, error $\epsilon$), there exists a function $f \in \mathcal{F}_{T,L}$ such that the iterates $\boldsymbol{x}_k$ of Alg. 2 satisfy $\lim_{k \to \infty} f(\boldsymbol{x}_k) - f(\boldsymbol{x}_*) > 1$.*

*3. Given $\varepsilon > 0$, if the fixed step size $\eta$ satisfies $\eta \geq \min\left\{\frac{16\sqrt{\varepsilon}}{\sqrt{T}L\sigma_*}, \frac{1}{TL}\right\}$, then there exists a function $f \in \mathcal{F}_{T,L}$ such that the iterates $\boldsymbol{x}_k$ of Alg. 2 satisfy $f(\boldsymbol{x}_k) - f(\boldsymbol{x}_*) > \varepsilon$ for all sufficiently large $k$.*

*Proof sketch.* For all parts of the proof, we consider 1-dimensional quadratics

$$f(x) = \frac{1}{T}\sum_{t=1}^T f_t(x), \quad \text{where } f_t(x) = \frac{L}{2}(x - \delta_t)^2 \ (t \in [T])$$

for $L > 0$ and $\{\delta_t\}_{t \in [T]} \subseteq \mathbb{R}$. It is immediate that $f(x)$ is minimized at $x_* = \frac{1}{T}\sum_{t=1}^T \delta_t$. In this case, Alg. 2 using a fixed step size $\eta$ performs closed-form updates on $f$, i.e., $x_{k+1} = \gamma^n x_k + (1 - \gamma)\sum_{t=1}^T \gamma^{T-t}\delta_t$, where $\gamma = \frac{1}{\eta L+1} \in (0,1)$. Given any initial point $x_0$, by iterating we have

$$x_k - x_* = \gamma^{kT}x_0 + \sum_{t=1}^T \left(\frac{\gamma^{T-t}(1-\gamma)(1-\gamma^{kT})}{1-\gamma^T} - \frac{1}{T}\right)\delta_t \xrightarrow{k\to\infty} \sum_{t=1}^T \left(\frac{\gamma^{T-t}(1-\gamma)}{1-\gamma^T} - \frac{1}{T}\right)\delta_t.$$

For 1), since $\gamma \in (0,1)$, $\frac{1-\gamma}{1-\gamma^T} - \frac{1}{T} = \frac{1}{\sum_{t=0}^{T-1}\gamma^t} - \frac{1}{T} > 0$ at $t = T$. As $k \to \infty$, for $\{\delta_t\}_{t\in[T]}$ with $\mathrm{sgn}(\delta_t) = \mathrm{sgn}\left(\frac{\gamma^{T-t}(1-\gamma)}{1-\gamma^T} - \frac{1}{T}\right)$, we have $f(x_k) - f(x_*) = \frac{L}{2}(x_k - x_*)^2 \geq \frac{L}{2}\left(\frac{1-\gamma}{1-\gamma^T} - \frac{1}{T}\right)^2 \delta_T^2 > 0$.

For 2), further observe that as $\gamma \in (0,1)$, $\left|\frac{\gamma^{T-t}(1-\gamma)}{1-\gamma^T} - \frac{1}{T}\right| \leq \frac{T-1}{T}$. As $k \to \infty$, for $|\delta_t| < \frac{T(2/L)^{1/2}}{(T-1)^2}$ ($t \in [T-1]$) and $\delta_T \geq \frac{2(2/L)^{1/2}}{\frac{1-\gamma}{1-\gamma^T} - \frac{1}{T}}$: $f(x_k) - f(x_*) > \frac{L}{2}\left(\left(\frac{1-\gamma}{1-\gamma^T} - \frac{1}{T}\right)\delta_T - \sum_{t=1}^{T-1}\frac{T-1}{T}|\delta_t|\right)^2 = 1$.

For 3), the case $\eta \geq \frac{1}{TL}$ can be handled similarly as in 1), so we assume w.l.o.g. that $\gamma \geq 1 - \frac{1}{T+1}$. Let $\gamma = 1 - \kappa$, where $0 < \kappa \leq \frac{1}{T+1} < \frac{1}{T}$. Further noticing that $(1-\kappa)^T \geq 1 - \kappa T + \frac{\kappa^2 T(T-1)}{4}$ for $T \geq 2$ and $\kappa T < 1$, then we have that the coefficient of $\delta_T$ is

$$\frac{1-\gamma}{1-\gamma^T} - \frac{1}{T} = \frac{\kappa}{1-(1-\kappa)^T} - \frac{1}{T} \geq \frac{\kappa}{\kappa T - \frac{\kappa^2 T(T-1)}{4}} - \frac{1}{T} \geq \frac{\kappa(T-1)}{4T} \geq \frac{\kappa}{8}.$$

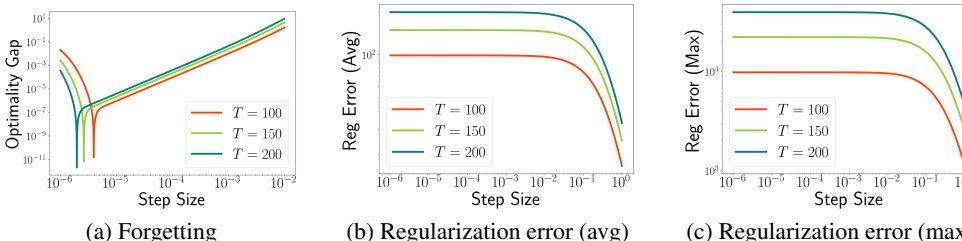

(a) Forgetting  (b) Regularization error (avg)  (c) Regularization error (max)

Figure 2: Numerical results for performing IPM on $T$ component least square functions, corresponding to the $\ell_2$-regularized CL setting with $T$ linear regression tasks with cyclic replays.

If $\eta \geq \frac{16\sqrt{\varepsilon}}{\sqrt{T}L\sigma_*}$, since $\sigma_*^2 \leq L^2 \sum_{t=1}^{T} \delta_t^2/T$ in this example, $\kappa = \frac{\eta L}{\eta L+1} \geq \frac{16\sqrt{\varepsilon/L}}{16\sqrt{\varepsilon/L}+\sqrt{\sum_{t=1}^{T}\delta_t^2}} > \frac{8\sqrt{3\varepsilon/L}}{\sqrt{\sum_{t=1}^{T}\delta_t^2}}$ with choosing large enough $\sum_{t=1}^{T}\delta_t^2$. Then for sufficiently large $k$ and $\{\delta_t\}_{t\in[T]}$ such that $\delta_T^2 > \frac{5}{6}\sum_{t=1}^{T}\delta_t^2$ and $\mathrm{sgn}(\delta_t) = \mathrm{sgn}\big(\frac{\gamma^{T-t}(1-\gamma)}{1-\gamma^T} - \frac{1}{T}\big)$: $f(x_k) - f(x_*) \geq \frac{2L}{5}\frac{3\varepsilon\delta_T^2}{L\sum_{t=1}^{T}\delta_t^2} > \varepsilon$. $\square$

**Regularization effect.** We now discuss how the regularization parameter (the inverse step size of IPM) affects the loss on the current task and (excess) forgetting, based on the above convergence results for IPM. An interesting aspect of our result in Theorem 2 is that there is a critical value

$$\eta_* = \min\left\{\frac{\|\boldsymbol{x}_0-\boldsymbol{x}_*\|^{2/3}}{2^{1/3}T\sigma_*^{2/3}L^{1/3}K^{1/3}(1+\beta/\alpha)^{1/3}}, \frac{1}{\sqrt{\beta}TL}\right\} \tag{7}$$

such that decreasing $\eta$ beyond $\eta_*$, both the regularization error on the current task and our upper bound on the forgetting increase. For $\eta > \eta_*$ (i.e., when we decrease the regularization error), Theorem 3 shows that the forgetting would increase, at least in some regimes of the problem parameters. Moreover, Theorem 3 demonstrates that polynomial dependence on other parameters like $\epsilon$ and $1/\sigma_*$ is necessary in the choice for $\eta$. In other words, strong regularization is needed to control the forgetting to a target error in general smooth convex settings. Another direct implication of Theorem 3 is that if no assumptions such as similarity are made on the tasks, then any $\ell_2$-regularized model using a finite regularization parameter would suffer catastrophic forgetting, i.e., the forgetting would not be approaching zero as the number of epochs tends to infinity.

We further provide illustrative numerical results in Fig. 2 to facilitate our discussion. In particular, we choose $L = 2$, $T \in \{100, 150, 200\}$, $\delta_t = 1/t$ ($t \in [T-1]$) and $\delta_T = T$ for the example $f(x) = \frac{L}{2T}\sum_{t=1}^{T}(x - \delta_t)^2$ used in the proof of Theorem 3. In Fig. 2(a), we plot the optimality gap at the last iterate, i.e., the excess forgetting, against the step sizes after $K = 10^4$ epochs. It can be observed that the forgetting first decreases with reducing the step size, but then increases beyond some critical value. Note that the critical values are around $10^{-5}$, which is nontrivially smaller than $1/L = 1/2$, while a larger $T$ leads to a smaller such critical value. These numerical examples corroborate our results from Theorems 2 and 3, which jointly suggest that the step size (amount of regularization) can neither be too small nor too large. On the other hand, in Fig. 2(b)-(c) we show the final stagnated average and maximum regularization error, i.e., $\frac{1}{T}\sum_{t=1}^{T} f_t(\boldsymbol{x}_{k-1,t+1}) - f_t(\boldsymbol{x}_{*,t})$ and $\max_{t\in[T]} f_t(\boldsymbol{x}_{k-1,t+1}) - f_t(\boldsymbol{x}_{*,t})$ over $T$ tasks, where $\boldsymbol{x}_{*,t}$ is the minimizer of $f_t$. We thus conclude from Fig. 2 that as the step size increases (equivalently, regularization parameter decreases), the regularization error decreases as well, but the forgetting increases.

To conclude, our results demonstrate that there is an inherent trade-off between forgetting and regularization error in CL with cyclic replaying of regularized smooth convex tasks. It is an interesting question for future research how incorporating task similarity into the analysis and even into the employed regularizing function might affect the conclusions.

### 3.2 CONVEX LIPSCHITZ SETTING

We now further relax the smoothness assumption and consider the convex Lipschitz setting, with applications such as linear classification tasks considered in Evron et al. (2023). To carry out the

analysis, we leverage the standard fact that the proximal iteration is equivalent to the gradient step with respect to the Moreau envelope, i.e.,

$$\boldsymbol{x}_{k-1,t+1} = \boldsymbol{x}_{k-1,t} - \eta_k \nabla M_{\eta_k f_t}(\boldsymbol{x}_{k-1,t}),$$

while the gradient of the Moreau envelope $\nabla M_{\eta_k f_t}(\boldsymbol{x}_{k-1,t})$ belongs to $\partial f_t(\boldsymbol{x}_{k-1,t+1})$, thus is bounded by the Lipschitz constant of $f_t$. We use these observations to bound the gap $f(\boldsymbol{x}_k) - f(\boldsymbol{z})$ for each epoch and then use the sequence $\{\boldsymbol{z}_k\}$ defined in Eq. (4) to deduce the last iterate rate. The results are summarized in the following theorem, while the proofs are deferred to Appendix C.

**Theorem 4.** *Under Assumptions 1 and 2, the output $\boldsymbol{x}_K$ of Alg. 2 satisfies*

$$f(\boldsymbol{x}_K) - f(\boldsymbol{x}_*) \le \frac{1}{2T\sum_{k=1}^K \eta_k} \|\boldsymbol{x}_0 - \boldsymbol{x}_*\|^2 + \frac{G^2 T}{2} \sum_{k=1}^K \frac{\eta_k^2}{\sum_{j=k}^K \eta_j}.$$

*Moreover, given $\epsilon > 0$, there exists a constant step size $\eta = \frac{\|\boldsymbol{x}_0 - \boldsymbol{x}_*\|}{GT\sqrt{K}}$ such that $f(\boldsymbol{x}_K) - f(\boldsymbol{x}_*) \le \epsilon$ after $\widetilde{\mathcal{O}}\big(\frac{G^2 T\|\boldsymbol{x}_0 - \boldsymbol{x}_*\|^2}{\epsilon^2}\big)$ individual proximal oracle queries.*

The last iterate rate we obtained in Theorem 4 matches the best known prior results for the average iterate of incremental proximal methods in nonsmooth settings (Bertsekas, 2011; Li et al., 2019), up to a logarithmic factor. Further, we take the $\Theta(\frac{1}{\sqrt{K}})$ step size only for analytical simplicity, while the diminishing step sizes $\eta_k = \Theta(\frac{1}{\sqrt{k}})$ will yield the same rate via a similar analysis.

## 3.3 Inexact proximal point evaluations

In the last two subsections, we derived our results assuming that the proximal point operator can be evaluated exactly. However, computing the proximal point corresponds to solving a strongly convex problem, which is generally possible to do only up to a finite accuracy. In the context of CL, it is also more realistic to assume that the loss of each task is minimized to certain accuracy, rather than exactly. Thus, we now consider the case where $\boldsymbol{x}_{k-1,t+1}$ is an approximation of $\text{prox}_{\eta_k f_t}(\boldsymbol{x}_{k-1,t})$ with solving the corresponding strongly convex problem to $\varepsilon_{k-1,t}^2/2\eta_k$-optimality gap for $\varepsilon_{k-1,t} > 0$. Equivalently, using strong convexity and denoting $\boldsymbol{g}_{k-1,t} := \frac{1}{\eta_k}(\boldsymbol{x}_{k-1,t} - \boldsymbol{x}_{k-1,t+1})$, we have

$$\|\boldsymbol{x}_{k-1,t+1} - \text{prox}_{\eta_k f_t}(\boldsymbol{x}_{k-1,t})\| \le \varepsilon_{k-1,t}, \quad \|\boldsymbol{g}_{k-1,t} - \nabla M_{\eta_k f_t}(\boldsymbol{x}_{k-1,t})\| \le \varepsilon_{k-1,t}/\eta_k. \quad (8)$$

We note that direct extensions of the previous analysis would not work, because inexact evaluations give rise to additional positive terms $\|\boldsymbol{x}_k - \boldsymbol{z}_{k-1}\|^2$ that cause issues for telescoping. However, we observe that the coefficients of these terms admit additional slackness, i.e., $(\lambda_k^2 w_k - w_{k-1})\|\boldsymbol{x}_k - \boldsymbol{z}_{k-1}\|^2$, while Lemma 2 only requires $\lambda_k w_k \le w_{k-1}$. Thus, as long as the approximation error at each iteration is small, we can still maintain the convergence rate of the IPM with exact proximal point evaluations. With these insights, we extend our convergence results to admit inexact proximal point evaluations as summarized in the following corollaries, with proofs in Appendix C.

**Corollary 3** (Convex Smooth). *Under Assumptions 1 and 3, 4 and for parameters $\alpha, \beta$ such that $\frac{1}{\alpha} + \frac{1}{\beta} \le \frac{1}{2}$, if the step size is fixed and satisfies $\eta_k \equiv \eta \le \frac{1}{\sqrt{\beta}TL}$, the output $\boldsymbol{x}_K$ of Alg. 2 with inexact proximal point evaluations as in Eq. (8) with $\sum_{t=1}^T \varepsilon_{k-1,t} \le \frac{\sqrt{\eta}}{1+(1+\alpha/\beta)(K-k+1)}$ satisfies*

$$f(\boldsymbol{x}_K) - f(\boldsymbol{x}_*) \le \frac{\text{e}\|\boldsymbol{x}_0 - \boldsymbol{x}_*\|^2}{2\eta TK^{\frac{1}{1+\alpha/\beta}}} + 2\eta^2 T^2 \sigma_*^2 L(1 + \beta/\alpha) K^{\frac{\alpha/\beta}{1+\alpha/\beta}} + \frac{\text{e}}{2\eta T} \sum_{k=0}^{K-1} \sum_{t=1}^T \frac{2T\varepsilon_{k,t}^2 + \sqrt{\eta}\varepsilon_{k,t}}{(K-k)^{\frac{1}{1+\alpha/\beta}}}.$$

*Given $\epsilon > 0$, if $\sum_{t=1}^T \varepsilon_{k-1,t} \le \sqrt{\eta}\min\{\varepsilon, \frac{1}{3(K-k+1)}\}$, there exists $\eta$ such that $f(\boldsymbol{x}_K) - f(\boldsymbol{x}_*) \le \epsilon$ after $\widetilde{\mathcal{O}}\big(\frac{TL\|\boldsymbol{x}_0 - \boldsymbol{x}_*\|^2}{\epsilon} + \frac{TL^{1/2}\sigma_*\|\boldsymbol{x}_0 - \boldsymbol{x}_*\|^2}{\epsilon^{3/2}}\big)$ individual inexact proximal point evaluations.*

**Corollary 4** (Convex Lipschitz). *Under Assumptions 1 and 2, the output $\boldsymbol{x}_K$ of Alg. 2 with inexact proximal point evaluations as in Eq. (8) with $\sum_{t=1}^T \varepsilon_{k-1,t} \le \frac{\eta_k \eta_{k-1} GT}{\sum_{j=k}^K \eta_j}$ satisfies*

$$f(\boldsymbol{x}_K) - f(\boldsymbol{x}_*) \le \frac{\|\boldsymbol{x}_0 - \boldsymbol{x}_*\|^2}{2T\sum_{k=1}^K \eta_k} + \frac{G^2 T}{2} \sum_{k=1}^K \frac{\eta_k^2}{\sum_{j=k}^K \eta_j} + \sum_{k=1}^K \sum_{t=1}^T \big(\frac{\varepsilon_{k-1,t}^2}{2T\sum_{j=k}^K \eta_j} + \frac{3G\varepsilon_{k-1,t}\eta_k}{\sum_{j=k}^K \eta_j}\big).$$

*Given $\epsilon > 0$, if $\sum_{t=1}^T \varepsilon_{k-1,t} \le \frac{2\eta GT}{K-k+1}$, there exists a constant step size $\eta$ such that $f(\boldsymbol{x}_K) - f(\boldsymbol{x}_*) \le \epsilon$ after $\widetilde{\mathcal{O}}\big(\frac{G^2 T\|\boldsymbol{x}_0 - \boldsymbol{x}_*\|^2}{\epsilon^2}\big)$ individual inexact proximal point evaluations.*

Due to space constraints, conclusions and discussions of future directions appear in Appendix D.

ACKNOWLEDGMENTS

This research was supported in part by the U.S. Office of Naval Research under contract number N00014-22-1-2348 and by the NSF CAREER Award CCF-2440563.

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

Table 1: Comparison of our results with state of the art, in terms of individual gradient/proximal oracle complexity required to output $\boldsymbol{x}_{\text{out}}$ with $f(\boldsymbol{x}_{\text{out}}) - f(\boldsymbol{x}_*) \leq \epsilon$ (IGD and IPM) or $\mathbb{E}[f(\boldsymbol{x}_{\text{out}}) - f(\boldsymbol{x}_*)] \leq \epsilon$ (RR/SO), where $\epsilon > 0$ is the target error and $\boldsymbol{x}_*$ is the optimal solution. Here, $\sigma_*^2 = \frac{1}{T}\sum_{t=1}^{T} \|\nabla f_t(\boldsymbol{x}_*)\|_2^2$, $D = \|\boldsymbol{x}_0 - \boldsymbol{x}_*\|_2$, and $\tilde{\mathcal{O}}(\cdot)$ suppresses up to square-root-log factors of $1/\varepsilon$.

| PAPER | | ASSUMPTION | AVERAGE ITERATE COMPLEXITY | LAST ITERATE COMPLEXITY |
|---|---|---|---|---|
| MISHCHENKO ET AL. (2020) NGUYEN ET AL. (2021) CAI ET AL. (2024) | (IGD) | $f_t$: $L$-SMOOTH, CONVEX | $\mathcal{O}\left(\frac{TLD^2}{\epsilon} + \frac{T\sqrt{L}\sigma_* D^2}{\epsilon^{3/2}}\right)$ | – |
| **[Ours]** | (IGD) | $f_t$: $L$-SMOOTH, CONVEX | $\mathcal{O}\left(\frac{TLD^2}{\epsilon} + \frac{T\sqrt{L}\sigma_* D^2}{\epsilon^{3/2}}\right)$ | $\tilde{\mathcal{O}}\left(\frac{TLD^2}{\epsilon} + \frac{T\sqrt{L}\sigma_* D^2}{\epsilon^{3/2}}\right)$ |
| MISHCHENKO ET AL. (2020) NGUYEN ET AL. (2021) CHA ET AL. (2023) CAI ET AL. (2024) | (RR/SO) | $f_t$: $L$-SMOOTH, CONVEX | $\mathcal{O}\left(\frac{TLD^2}{\epsilon} + \frac{\sqrt{TL}\sigma_* D^2}{\epsilon^{3/2}}\right)$ | – |
| **[Ours]** | (RR/SO) | $f_t$: $L$-SMOOTH, CONVEX | $\mathcal{O}\left(\frac{TLD^2}{\epsilon} + \frac{\sqrt{TL}\sigma_* D^2}{\epsilon^{3/2}}\right)$ | $\tilde{\mathcal{O}}\left(\frac{TLD^2}{\epsilon} + \frac{\sqrt{TL}\sigma_* D^2}{\epsilon^{3/2}}\right)$ |
| BERTSEKAS (2011) LI ET AL. (2019) | (IPM) | $f_t$: $G$-LIPSCHITZ, CONVEX | $\mathcal{O}\left(\frac{TG^2 D^2}{\epsilon^2}\right)$ | – |
| **[Ours]** | (IPM) | $f_t$: $L$-SMOOTH, CONVEX | $\mathcal{O}\left(\frac{TLD^2}{\epsilon} + \frac{T\sqrt{L}\sigma_* D^2}{\epsilon^{3/2}}\right)$ | $\tilde{\mathcal{O}}\left(\frac{TLD^2}{\epsilon} + \frac{T\sqrt{L}\sigma_* D^2}{\epsilon^{3/2}}\right)$ |
| **[Ours]** | (IPM) | $f_t$: $G$-LIPSCHITZ, CONVEX | $\mathcal{O}\left(\frac{TG^2 D^2}{\epsilon^2}\right)$ | $\tilde{\mathcal{O}}\left(\frac{TG^2 D^2}{\epsilon^2}\right)$ |

## A RELATED WORK

We first provide Table 1 for a comparison of our results on incremental methods with the state-of-the-art on average iterate convergence.

To remedy catastrophic forgetting, various empirical approaches have been proposed, corresponding to three main categories: (i) memory-based approaches, which store samples from previous tasks and reuse those data for training on the current task (Robins, 1995; Lopez-Paz & Ranzato, 2017; Rolnick et al., 2019); (ii) expansion-based approaches, which progressively expand the model size to accommodate new tasks (Rusu et al., 2016; Yoon et al., 2017); and (iii) regularization-based approaches, which regularize the loss of the current task to ensure the new model parameter vector is close to the old ones (Kirkpatrick et al., 2017); for a more complete survey, we refer to De Lange et al. (2021). Our focus in this work is on (i) and (iii).

On the theoretical front, the majority of existing work for the settings we consider has studied catastrophic forgetting for overparameterized linear models (Evron et al., 2022; 2023; Lin et al., 2023b; Goldfarb & Hand, 2023; Goldfarb et al., 2024; Swartworth et al., 2024) or under the neural tangent kernel regime (Bennani et al., 2020; Doan et al., 2021). Another related line of work studied the task similarity in CL under the teacher-student learning framework (Asanuma et al., 2021; Lee et al., 2021). A recent work showed that an algorithm recursively minimizing the loss of the current task over the solution space of the previous task, named the ideal continual learner, never forgets, assuming all tasks share a common global minimizer (Peng et al., 2023).

When it comes to the cyclic forgetting that this paper focuses on, theoretical results have only been established for linear models (Evron et al., 2022; 2023; Swartworth et al., 2024). In particular, the analysis for linear regression tasks in Evron et al. (2022); Swartworth et al. (2024) crucially relies on the exact minimization of each task using (S)GD to have closed-form updates between tasks, while Evron et al. (2023) uses alternating projections to analyze linear classification tasks. It is unclear how to extend either of these results to general convex loss functions that we consider.

On a technical level, our results are most closely related to 1) the literature on last iterate convergence guarantees for subgradient-based methods and stochastic gradient descent (SGD) and 2) the

literature on incremental gradient methods and shuffled SGD. For 1), we draw inspiration from the recent results (Zamani & Glineur, 2023; Liu & Zhou, 2024a), which rely on a clever construction of reference points $z_k$ with respect to which a gap quantity $f(x_k) - f(z_{k-1})$ gets bounded to deduce a bound for the optimality gap $f(x_k) - f(x_*)$ of the last iterate $x_*$. For the latter line of work 2), we generalize the analysis used exclusively for the optimality gap of the (uniformly) average iterate (Mishchenko et al., 2020; Nguyen et al., 2021; Cha et al., 2023; Cai et al., 2024) to control the gap-like quantities $f(x_k) - f(z_{k-1})$, which require a more careful argument for controlling all error terms introduced by replacing $x_*$ by $z_{k-1}$ without introducing spurious unrealistic assumptions about the magnitudes of the component functions' gradients.

For further comparison with the works on last iterate guarantees (Zamani & Glineur, 2023; Liu & Zhou, 2024a), we note that Zamani & Glineur (2023) study subgradient methods for a single deterministic objective function in convex Lipschitz settings. Since the work is not dealing with finite sum objectives and incremental updates, the focus is different and the algorithms are not directly comparable, barring for trivial cases (e.g., setting $T = 1$ in our results). Further, Liu & Zhou (2024a) study the last-iterate convergence for classical SGD with replacement sampling, in stochastic convex settings. The major challenge in our setting is that the functions are accessed in a cyclic order, as opposed to being randomly sampled as in Liu & Zhou (2024a). We note that analyzing SGD with cyclic ordering of component gradients (as opposed to standard SGD that uses random sampling with replacement) was a major open problem for decades (Bottou, 2009), while existing results rely on a completely different analysis. For additional results on the last iterate convergence of classical SGD, which is less relevant to this work, we direct the reader to the references in Jain et al. (2021); Liu & Zhou (2024a).

We finally note that both these related lines of work concern problems on which progress was made only in the very recent literature. In particular, while the oracle complexity upper bound for the average iterate of SGD in convex Lipschitz-continuous settings has been known for decades and its analysis is routinely taught in optimization and machine learning classes, there were no such results for the last iterate of SGD until 2013 (Shamir & Zhang, 2013) with improvements and generalizations to these results obtained as recently as in the past several months (Liu & Zhou, 2024a). Regarding 2), obtaining any nonasymptotic convergence guarantees for incremental gradient methods/shuffled SGD had remained open for decades (Bottou, 2009) until a recent line of work (Gürbüzbalaban et al., 2021; Shamir, 2016; Haochen & Sra, 2019; Nagaraj et al., 2019; Ahn et al., 2020; Rajput et al., 2020; Yun et al., 2022; Safran & Shamir, 2020). For nonconvex problems, Yu & Li (2023) proposed a stopping criteria for shuffled SGD and proved the high-probability guarantee for the exiting point, which is technically disjoint from our work. For smooth convex problems we consider, the convergence results were obtained only in the past few years (Mishchenko et al., 2020; Nguyen et al., 2021; Cha et al., 2023) and improved in Cai et al. (2024) using a fine-grained analysis inspired by the recent advances in cyclic methods (Song & Diakonikolas, 2023; Cai et al., 2023; Lin et al., 2023a). However, all those results are for the (uniformly) average iterate, while obtaining convergence results for the last iterate had remained open until this work.

An independent and concurrent work to ours (Liu & Zhou, 2024b) studied the last-iterate convergence of shuffled SGD for composite (strongly) convex smooth/Lipschitz optimization. For the same problems as studied in our Section 2, they obtained the same convergence results. The remaining results in Liu & Zhou (2024b) and our work are not directly comparable, as the motivation for the two works and the studied settings are different: we do not consider composite optimization settings for shuffled SGD/incremental gradient nor strong convexity of component functions, while Liu & Zhou (2024b) did not consider the incremental proximal method (our Section 3). It is of note that while Liu & Zhou (2024b) used proximal steps to handle the nonsmooth portion of the objective in their composite setting, the proximal maps are not applied component-wise, but at the end of a cycle, to only one (regularizer) function.

**Further discussion on other related topics.** In the context of more general finite-sum minimization, one interesting application of our results is in federated learning, where most of the existing results are proved for the average iterate; see e.g., Alistarh et al. (2017). Last-iterate results in federated learning have only been obtained for different algorithms without incremental updates over component gradients, see e.g., Li (2021); Li & Richtárik (2021). Hence, our last-iterate results are still new, even to the field of federated learning.

Incremental methods this work focuses on also have a natural connection to cyclic coordinate methods. However, the results for cyclic coordinate methods are not directly applicable to deriving convergence guarantees for incremental methods. For the direct reformulation of finite sum minimization to $\min_{\boldsymbol{x}, z_1, \dots, z_T} \frac{1}{T} \sum_{t=1}^{T} z_t$, s.t. $z_t = f_t(\boldsymbol{x})$ ($t \in [T]$) with introducing additional variables, one needs to handle nonseparable constraints associated with generic functions $z_t = f_t(\boldsymbol{x})$. These constraints are nonconvex unless $f_t$'s are linear. In most works on cyclic block coordinate methods for composite optimization (see e.g., Cai et al. (2023) and references therein), the regularizer is assumed to be convex and generally cannot enforce such hard constraints. It is worth noting that Chorobura & Necoara (2023) derive convergence results for the last iterate of a cyclic coordinate method applied to problems with nonseparable constraints. However, Chorobura & Necoara (2023) require additional assumptions on the regularizing function, such as either being weakly convex with easy proximal steps or being concave along coordinates with additional assumptions made about its Hessian. Their last iterate result also depends on a uniform bound over the iterate distance to optima, which would be ill-defined if ported to the setting of incremental methods.

# B  OMITTED PROOFS FROM SECTION 2

**Lemma 1.** *Under Assumptions 1 and 3, for any $\boldsymbol{z} \in \mathbb{R}^d$ that is fixed in the $k$-th cycle of Alg. 1 and any $\alpha, \beta > 0$ such that $\frac{1}{\alpha} + \frac{1}{\beta} \leq \frac{1}{2}$, if $\eta_k \leq \frac{1}{\sqrt{\beta}TL}$, then for all $k \in [K]$,*

$$T\big(f(\boldsymbol{x}_k) - f(\boldsymbol{z})\big) \leq \eta_k^2 L \sum_{t=1}^{T} \big\| \sum_{s=t}^{T} \nabla f_s(\boldsymbol{x}_*) \big\|^2 + \tfrac{\alpha}{\beta} T\big(f(\boldsymbol{z}) - f(\boldsymbol{x}_*)\big)$$
$$+ \tfrac{1}{2\eta_k}\big(\|\boldsymbol{x}_{k-1} - \boldsymbol{z}\|^2 - \|\boldsymbol{x}_k - \boldsymbol{z}\|^2\big).$$

*Proof.* Since each $f_t$ is convex and $L$-smooth, we have for $t \in [T]$ (see Eq. (3)):

$$f_t(\boldsymbol{x}_k) - f_t(\boldsymbol{x}_{k-1,t}) \leq \langle \nabla f_t(\boldsymbol{x}_k), \boldsymbol{x}_k - \boldsymbol{x}_{k-1,t}\rangle - \frac{1}{2L}\|\nabla f_t(\boldsymbol{x}_k) - \nabla f_t(\boldsymbol{x}_{k-1,t})\|^2, \qquad (9)$$

$$f_t(\boldsymbol{x}_{k-1,t}) - f_t(\boldsymbol{z}) \leq \langle \nabla f_t(\boldsymbol{x}_{k-1,t}), \boldsymbol{x}_{k-1,t} - \boldsymbol{z}\rangle - \frac{1}{2L}\|\nabla f_t(\boldsymbol{x}_{k-1,t}) - \nabla f_t(\boldsymbol{z})\|^2. \qquad (10)$$

On the other hand, letting $\Phi_{k-1,t}(\boldsymbol{x}) := \langle \nabla f_t(\boldsymbol{x}_{k-1,t}), \boldsymbol{x}\rangle + \frac{1}{2\eta_k}\|\boldsymbol{x}_{k-1,t} - \boldsymbol{x}\|^2$, we have $\nabla \Phi_{k-1,t}(\boldsymbol{x}_{k-1,t+1}) = \boldsymbol{0}$ by the update $\boldsymbol{x}_{k-1,t+1} = \boldsymbol{x}_{k-1,t} - \eta_k \nabla f_t(\boldsymbol{x}_{k-1,t})$ in Alg. 1. Observe that $\Phi_{k-1,t}$ is $\frac{1}{\eta_k}$-strong convex, and thus we also have

$$\Phi_{k-1,t}(\boldsymbol{z}) \geq \Phi_{k-1,t}(\boldsymbol{x}_{k-1,t+1}) + \frac{1}{2\eta_k}\|\boldsymbol{z} - \boldsymbol{x}_{k-1,t+1}\|^2. \qquad (11)$$

Summing Eq. (9) and (10) and using Eq. (11), we conclude that for $t \in [T]$,

$$f_t(\boldsymbol{x}_k) - f_t(\boldsymbol{z}) \leq \langle \nabla f_t(\boldsymbol{x}_k), \boldsymbol{x}_k - \boldsymbol{x}_{k-1,t}\rangle + \langle \nabla f_t(\boldsymbol{x}_{k-1,t}), \boldsymbol{x}_{k-1,t} - \boldsymbol{x}_{k-1,t+1}\rangle$$
$$- \frac{1}{2L}\|\nabla f_t(\boldsymbol{x}_k) - \nabla f_t(\boldsymbol{x}_{k-1,t})\|^2 - \frac{1}{2L}\|\nabla f_t(\boldsymbol{x}_{k-1,t}) - \nabla f_t(\boldsymbol{z})\|^2$$
$$- \frac{1}{2\eta_k}\|\boldsymbol{x}_{k-1,t+1} - \boldsymbol{x}_{k-1,t}\|^2 + \frac{1}{2\eta_k}\big(\|\boldsymbol{x}_{k-1,t} - \boldsymbol{z}\|^2 - \|\boldsymbol{x}_{k-1,t+1} - \boldsymbol{z}\|^2\big).$$

Decomposing $\nabla f_t(\boldsymbol{x}_k) = \nabla f_t(\boldsymbol{x}_k) - \nabla f_t(\boldsymbol{x}_{k-1,t}) + \nabla f_t(\boldsymbol{x}_{k-1,t})$ and summing over $t \in [T]$ where we recall $\boldsymbol{x}_{k-1} = \boldsymbol{x}_{k-1,1}$ and $\boldsymbol{x}_k = \boldsymbol{x}_{k-1,T+1}$, we have

$$T\big(f(\boldsymbol{x}_k) - f(\boldsymbol{z})\big) \le \underbrace{\sum_{t=1}^{T} \langle \nabla f_t(\boldsymbol{x}_{k-1,t}), \boldsymbol{x}_k - \boldsymbol{x}_{k-1,t+1} \rangle - \frac{1}{2\eta_k} \sum_{t=1}^{T} \|\boldsymbol{x}_{k-1,t+1} - \boldsymbol{x}_{k-1,t}\|^2}_{\mathcal{T}_1}$$

$$+ \underbrace{\sum_{t=1}^{T} \langle \nabla f_t(\boldsymbol{x}_k) - \nabla f_t(\boldsymbol{x}_{k-1,t}), \boldsymbol{x}_k - \boldsymbol{x}_{k-1,t} \rangle}_{\mathcal{T}_2}$$

$$- \frac{1}{2L} \sum_{t=1}^{T} \|\nabla f_t(\boldsymbol{x}_k) - \nabla f_t(\boldsymbol{x}_{k-1,t})\|^2 - \frac{1}{2L} \sum_{t=1}^{T} \|\nabla f_t(\boldsymbol{x}_{k-1,t}) - \nabla f_t(\boldsymbol{z})\|^2$$

$$+ \frac{1}{2\eta_k} \big(\|\boldsymbol{x}_{k-1} - \boldsymbol{z}\|^2 - \|\boldsymbol{x}_k - \boldsymbol{z}\|^2\big).$$

For the term $\mathcal{T}_1$, we recall that by the IGD update, $\nabla f_t(\boldsymbol{x}_{k-1,t}) = -\frac{1}{\eta_k}(\boldsymbol{x}_{k-1,t+1} - \boldsymbol{x}_{k-1,t})$ and $\boldsymbol{x}_k - \boldsymbol{x}_{k-1,t+1} = \sum_{s=t+1}^{T}(\boldsymbol{x}_{k-1,s+1} - \boldsymbol{x}_{k-1,s})$, and thus we have

$$\mathcal{T}_1 = -\frac{1}{\eta_k} \sum_{t=1}^{T-1} \sum_{s=t+1}^{T} \langle \boldsymbol{x}_{k-1,t+1} - \boldsymbol{x}_{k-1,t}, \boldsymbol{x}_{k-1,s+1} - \boldsymbol{x}_{k-1,s} \rangle - \frac{1}{2\eta_k} \sum_{t=1}^{T} \|\boldsymbol{x}_{k-1,t+1} - \boldsymbol{x}_{k-1,t}\|^2$$

$$= -\frac{1}{2\eta_k} \Big\| \sum_{t=1}^{T} (\boldsymbol{x}_{k-1,t+1} - \boldsymbol{x}_{k-1,t}) \Big\|^2 = -\frac{1}{2\eta_k} \|\boldsymbol{x}_k - \boldsymbol{x}_{k-1}\|^2 \le 0.$$

For the term $\mathcal{T}_2$, noticing that $\boldsymbol{x}_k - \boldsymbol{x}_{k-1,t} = -\eta_k \sum_{s=t}^{T} \nabla f_s(\boldsymbol{x}_{k-1,s})$ and decomposing $\nabla f_s(\boldsymbol{x}_{k-1,s}) = (\nabla f_s(\boldsymbol{x}_{k-1,s}) - \nabla f_s(\boldsymbol{z})) + (\nabla f_s(\boldsymbol{z}) - \nabla f_s(\boldsymbol{x}_*)) + \nabla f_s(\boldsymbol{x}_*)$, we use Young's inequality with parameters $\alpha > 0$ and $\beta > 0$ to obtain

$$\mathcal{T}_2 = \sum_{t=1}^{T} \Big\langle \nabla f_t(\boldsymbol{x}_k) - \nabla f_t(\boldsymbol{x}_{k-1,t}), -\eta_k \sum_{s=t}^{T} \nabla f_s(\boldsymbol{x}_{k-1,s}) \Big\rangle$$

$$\le \frac{1}{2L} \Big(\frac{1}{2} + \frac{1}{\alpha} + \frac{1}{\beta}\Big) \sum_{t=1}^{T} \|\nabla f_t(\boldsymbol{x}_k) - \nabla f_t(\boldsymbol{x}_{k-1,t})\|^2$$

$$+ \frac{\alpha \eta_k^2 L}{2} \sum_{t=1}^{T} \Big\| \sum_{s=t}^{T} (\nabla f_s(\boldsymbol{z}) - \nabla f_s(\boldsymbol{x}_*)) \Big\|^2 + \eta_k^2 L \sum_{t=1}^{T} \Big\| \sum_{s=t}^{T} \nabla f_s(\boldsymbol{x}_*) \Big\|^2$$

$$+ \frac{\beta \eta_k^2 L}{2} \sum_{t=1}^{T} \Big\| \sum_{s=t}^{T} (\nabla f_s(\boldsymbol{x}_{k-1,s}) - \nabla f_s(\boldsymbol{z})) \Big\|^2.$$

Further using that $\|\sum_{i=1}^{n} \boldsymbol{x}_i\|^2 \le n \sum_{i=1}^{n} \|\boldsymbol{x}_i\|^2$ and combining the above bounds on $\mathcal{T}_1$ and $\mathcal{T}_2$, we obtain

$$T\big(f(\boldsymbol{x}_k) - f(\boldsymbol{z})\big) \le \frac{1}{2L} \Big(\frac{1}{\alpha} + \frac{1}{\beta} - \frac{1}{2}\Big) \sum_{t=1}^{T} \|\nabla f_t(\boldsymbol{x}_k) - \nabla f_t(\boldsymbol{x}_{k-1,t})\|^2$$

$$+ \Big(\frac{\beta \eta_k^2 T^2 L}{2} - \frac{1}{2L}\Big) \sum_{t=1}^{T} \|\nabla f_t(\boldsymbol{x}_{k-1,t}) - \nabla f_t(\boldsymbol{z})\|^2$$

$$+ \frac{\alpha \eta_k^2 T^2 L}{2} \sum_{t=1}^{T} \|\nabla f_t(\boldsymbol{z}) - \nabla f_t(\boldsymbol{x}_*)\|^2 + \eta_k^2 L \sum_{t=1}^{T} \Big\| \sum_{s=t}^{T} \nabla f_s(\boldsymbol{x}_*) \Big\|^2$$

$$+ \frac{1}{2\eta_k} \big(\|\boldsymbol{x}_{k-1} - \boldsymbol{z}\|^2 - \|\boldsymbol{x}_k - \boldsymbol{z}\|^2\big),$$

To make the first two terms on the right-hand side both nonpositive, we choose

$$\frac{1}{\alpha} + \frac{1}{\beta} \le \frac{1}{2}, \quad \eta_k \le \frac{1}{\sqrt{\beta}TL}.$$

We further bound the term $\sum_{t=1}^{T} \|\nabla f_t(\boldsymbol{z}) - \nabla f_t(\boldsymbol{x}_*)\|^2$ by the smoothness and convexity of each $f_t$ as follows:

$$\sum_{t=1}^{T} \|\nabla f_t(\boldsymbol{z}) - \nabla f_t(\boldsymbol{x}_*)\|^2 \le 2L \sum_{t=1}^{T} \big( f_t(\boldsymbol{z}) - f_t(\boldsymbol{x}_*) - \langle \nabla f_t(\boldsymbol{x}_*), \boldsymbol{z} - \boldsymbol{x}_* \rangle \big)$$
$$= 2TL \big( f(\boldsymbol{z}) - f(\boldsymbol{x}_*) \big),$$

where in the last equation we used $\nabla f(\boldsymbol{x}_*) = \boldsymbol{0}$. Hence, we finally obtain

$$T\big(f(\boldsymbol{x}_k) - f(\boldsymbol{z})\big) \le \eta_k^2 L \sum_{t=1}^{T} \Big\| \sum_{s=t}^{T} \nabla f_s(\boldsymbol{x}_*) \Big\|^2 + \alpha T^3 \eta_k^2 L^2 \big( f(\boldsymbol{z}) - f(\boldsymbol{x}_*) \big)$$
$$+ \frac{1}{2\eta_k} \big( \|\boldsymbol{x}_{k-1} - \boldsymbol{z}\|^2 - \|\boldsymbol{x}_k - \boldsymbol{z}\|^2 \big)$$
$$\le \eta_k^2 L \sum_{t=1}^{T} \Big\| \sum_{s=t}^{T} \nabla f_s(\boldsymbol{x}_*) \Big\|^2 + \frac{\alpha}{\beta} T \big( f(\boldsymbol{z}) - f(\boldsymbol{x}_*) \big)$$
$$+ \frac{1}{2\eta_k} \big( \|\boldsymbol{x}_{k-1} - \boldsymbol{z}\|^2 - \|\boldsymbol{x}_k - \boldsymbol{z}\|^2 \big),$$

where we used $\eta_k \le \frac{1}{\sqrt{\beta}TL}$ in the last inequality, thus completing the proof. $\qquad\square$

**Lemma 2.** *Let $\boldsymbol{z}_k$ be defined by Eq. (4) for a given sequence of parameters $\lambda_k \in (0,1)$, where $k \ge 0$ and $\boldsymbol{z}_{-1} = \boldsymbol{x}_*$. Under Assumption 1, if there exists a sequence of nonnegative weights $w_k$ such that $\lambda_k w_k \le w_{k-1}$ for $k \in [K-1]$, then for all $k \in [K]$:*

*1. $w_{k-1}\big(f(\boldsymbol{z}_{k-1}) - f(\boldsymbol{x}_*)\big) \le \sum_{j=0}^{k-1} w_j (1 - \lambda_j)\big(f(\boldsymbol{x}_j) - f(\boldsymbol{x}_*)\big);$*

*2. $w_{k-1}\big(f(\boldsymbol{x}_k) - f(\boldsymbol{z}_{k-1})\big) \ge w_{k-1}(f(\boldsymbol{x}_k) - f(\boldsymbol{x}_*)) - \sum_{j=0}^{k-1} w_j (1 - \lambda_j)(f(\boldsymbol{x}_j) - f(\boldsymbol{x}_*)).$*

*Proof.*

1. Using Eq. (5) and the convexity of $f$, we have

$$f(\boldsymbol{z}_{k-1}) - f(\boldsymbol{x}_*)$$
$$\le (1 - \lambda_{k-1})\big(f(\boldsymbol{x}_{k-1}) - f(\boldsymbol{x}_*)\big) + \sum_{j=0}^{k-2} \Big( \prod_{i=j+1}^{k-1} \lambda_i \Big)(1 - \lambda_j)\big(f(\boldsymbol{x}_j) - f(\boldsymbol{x}_*)\big).$$

It remains to multiply by $w_{k-1}$ on both sides and notice that by the lemma assumption,

$$w_{k-1}\Big( \prod_{i=j+1}^{k-1} \lambda_i \Big)(1 - \lambda_j) \le w_{k-2}\Big( \prod_{i=j+1}^{k-2} \lambda_i \Big)(1 - \lambda_j) \le \cdots \le w_j(1 - \lambda_j).$$

2. This follows from the first part of the lemma, by decomposing $f(\boldsymbol{x}_k) - f(\boldsymbol{z}_{k-1}) = f(\boldsymbol{x}_k) - f(\boldsymbol{x}_*) - \big(f(\boldsymbol{z}_{k-1}) - f(\boldsymbol{x}_*)\big)$.

$$\square$$

**Theorem 1.** *Under Assumptions 1, 3, and 4 and for positive parameters $\alpha, \beta > 0$ such that $\frac{1}{\alpha} + \frac{1}{\beta} \le \frac{1}{2}$, if the step size is fixed and satisfies $\eta_k = \eta \le \frac{1}{\sqrt{\beta}TL}$, the output $\boldsymbol{x}_K$ of Alg. 1 satisfies*

$$f(\boldsymbol{x}_K) - f(\boldsymbol{x}_*) \le \mathrm{e}\eta^2 T^2 \sigma_*^2 L(1 + \beta/\alpha) K^{\frac{\alpha/\beta}{1+\alpha/\beta}} + \frac{\mathrm{e}\|\boldsymbol{x}_0 - \boldsymbol{x}_*\|^2}{2\eta T K^{\frac{1}{1+\alpha/\beta}}}. \tag{6}$$

*In particular, for $\alpha = 4, \beta = 4\log K$, and $\eta = \min\big\{\frac{\|\boldsymbol{x}_0 - \boldsymbol{x}_*\|^{2/3}}{2^{1/3}T\sigma_*^{2/3}L^{1/3}K^{1/3}(1+\log K)^{1/3}}, \frac{1}{2\sqrt{\log K}TL}\big\}$,*
*$f(\boldsymbol{x}_K) - f(\boldsymbol{x}_*) = \mathcal{O}\big(\frac{L^{1/3}\sigma_*^{2/3}\|\boldsymbol{x}_0 - \boldsymbol{x}_*\|^{4/3}(1+\log K)^{1/3}}{K^{2/3}} + \frac{L\|\boldsymbol{x}_0 - \boldsymbol{x}_*\|^2\sqrt{\log K}}{K}\big)$, and thus $f(\boldsymbol{x}_K) -$*
*$f(\boldsymbol{x}_*) \le \epsilon$ after $\widetilde{\mathcal{O}}\big(\frac{TL\|\boldsymbol{x}_0 - \boldsymbol{x}_*\|^2}{\epsilon} + \frac{TL^{1/2}\sigma_*\|\boldsymbol{x}_0 - \boldsymbol{x}_*\|^2}{\epsilon^{3/2}}\big)$ individual gradient evaluations.*

*Proof.* Plugging $\boldsymbol{z}_{k-1}$ defined by Eq. (4) into Lemma 1 and using the inequality $\|\sum_{i=1}^n \boldsymbol{x}_i\|^2 \le n\sum_{i=1}^n \|\boldsymbol{x}_i\|^2$ to bound the term $\sum_{t=1}^T \|\sum_{s=t}^T \nabla f_s(\boldsymbol{x}_*)\|^2$, we obtain

$$T\big(f(\boldsymbol{x}_k) - f(\boldsymbol{z}_{k-1})\big) \le T^3\eta_k^2\sigma_*^2 L + \frac{\alpha}{\beta}T\big(f(\boldsymbol{z}_{k-1}) - f(\boldsymbol{x}_*)\big)$$
$$+ \frac{1}{2\eta_k}\big(\|\boldsymbol{x}_{k-1} - \boldsymbol{z}_{k-1}\|^2 - \|\boldsymbol{x}_k - \boldsymbol{z}_{k-1}\|^2\big),$$

where $\frac{1}{\alpha} + \frac{1}{\beta} \le \frac{1}{2}$. Multiplying $\eta_k w_{k-1}$ on both sides with $w_k$ such that $\lambda_k w_k \le w_{k-1}$ and noticing that $\|\boldsymbol{x}_{k-1} - \boldsymbol{z}_{k-1}\|^2 \le \lambda_{k-1}\|\boldsymbol{x}_{k-1} - \boldsymbol{z}_{k-2}\|^2$ by Eq. (4), we have

$$T\eta_k w_{k-1}\big(f(\boldsymbol{x}_k) - f(\boldsymbol{z}_{k-1})\big) \le T^3\eta_k^3\sigma_*^2 L w_{k-1} + \frac{\alpha}{\beta}T\eta_k w_{k-1}\big(f(\boldsymbol{z}_{k-1}) - f(\boldsymbol{x}_*)\big)$$
$$+ \frac{1}{2}\big(w_{k-2}\|\boldsymbol{x}_{k-1} - \boldsymbol{z}_{k-2}\|^2 - w_{k-1}\|\boldsymbol{x}_k - \boldsymbol{z}_{k-1}\|^2\big)$$

We then sum over $k \in [K]$ and use Lemma 2 to obtain

$$T\sum_{k=1}^K \eta_k\Big[w_{k-1}\big(f(\boldsymbol{x}_k) - f(\boldsymbol{x}_*)\big) - \sum_{j=0}^{k-1} w_j(1 - \lambda_j)\big(f(\boldsymbol{x}_j) - f(\boldsymbol{x}_*)\big)\Big]$$
$$\le T^3\sigma_*^2 L\sum_{k=1}^K \eta_k^3 w_{k-1} + \frac{w_{-1}}{2}\|\boldsymbol{x}_0 - \boldsymbol{x}_*\|^2 + \frac{\alpha}{\beta}T\sum_{k=1}^K \eta_k \sum_{j=0}^{k-1} w_j(1 - \lambda_j)\big(f(\boldsymbol{x}_j) - f(\boldsymbol{x}_*)\big),$$
(12)

where we also use $\boldsymbol{z}_{-1} = \boldsymbol{x}_*$. Unrolling the terms w.r.t. $f(\boldsymbol{x}_k) - f(\boldsymbol{x}_*)$ $(k \in [K])$ we get

$$\sum_{k=1}^K \eta_k\Big[w_{k-1}\big(f(\boldsymbol{x}_k) - f(\boldsymbol{x}_*)\big) - \sum_{j=0}^{k-1} w_j(1 - \lambda_j)\big(f(\boldsymbol{x}_j) - f(\boldsymbol{x}_*)\big)\Big]$$
$$= \eta_K w_{K-1}\big(f(\boldsymbol{x}_K) - f(\boldsymbol{x}_*)\big) - w_0(1 - \lambda_0)\big(f(\boldsymbol{x}_0) - f(\boldsymbol{x}_*)\big)\sum_{k=1}^K \eta_k$$
(13)
$$+ \sum_{k=1}^{K-1}\Big(\eta_k w_{k-1} - w_k(1 - \lambda_k)\sum_{j=k+1}^K \eta_j\Big)\big(f(\boldsymbol{x}_k) - f(\boldsymbol{x}_*)\big)$$

and

$$\sum_{k=1}^K \eta_k \sum_{j=0}^{k-1} w_j(1 - \lambda_j)\big(f(\boldsymbol{x}_j) - f(\boldsymbol{x}_*)\big) = \sum_{k=0}^{K-1}\Big(w_k(1 - \lambda_k)\sum_{j=k+1}^K \eta_j\Big)\big(f(\boldsymbol{x}_k) - f(\boldsymbol{x}_*)\big).$$

Plugging back into Eq. (12), grouping the like terms, and choosing $\lambda_0 = 1$, we obtain

$$T\eta_K w_{K-1}\big(f(\boldsymbol{x}_K) - f(\boldsymbol{x}_*)\big)$$
$$+ T\sum_{k=1}^{K-1}\Big[\eta_k w_{k-1} - \big(1 + \frac{\alpha}{\beta}\big)w_k(1 - \lambda_k)\sum_{j=k+1}^K \eta_j\Big]\big(f(\boldsymbol{x}_k) - f(\boldsymbol{x}_*)\big)$$
(14)
$$\le T^3\sigma_*^2 L\sum_{k=1}^K \eta_k^3 w_{k-1} + \frac{w_{-1}}{2}\|\boldsymbol{x}_0 - \boldsymbol{x}_*\|^2.$$

To obtain the last iterate guarantee, it suffices to choose $w_k$ and $\lambda_k$ such that

$$\lambda_k w_k \le w_{k-1}, \quad 0 \le k \le K-1,$$
(15)

$$\eta_k w_{k-1} - \big(1 + \frac{\alpha}{\beta}\big)w_k(1 - \lambda_k)\sum_{j=k+1}^K \eta_j \ge 0, \quad 1 \le k \le K-1.$$
(16)

Noticing that Eq. 16 is equivalent to $\lambda_k \geq 1 - \frac{\eta_k w_{k-1}}{\left(1+\frac{\alpha}{\beta}\right) w_k \sum_{j=k+1}^K \eta_j}$, to have both inequalities satisfied at the same time, it suffices that

$$1 - \frac{\eta_k w_{k-1}}{\left(1+\frac{\alpha}{\beta}\right) w_k \sum_{j=k+1}^K \eta_j} \leq \frac{w_{k-1}}{w_k} \iff w_k \leq \frac{\eta_k + \left(1+\frac{\alpha}{\beta}\right) \sum_{j=k+1}^K \eta_j}{\left(1+\frac{\alpha}{\beta}\right) \sum_{j=k+1}^K \eta_j} w_{k-1}.$$

To maximize the growth rate of $\{w_k\}$, we let $w_k = \frac{\eta_k + \left(1+\frac{\alpha}{\beta}\right) \sum_{j=k+1}^K \eta_j}{\left(1+\frac{\alpha}{\beta}\right) \sum_{j=k+1}^K \eta_j} w_{k-1}$. Without loss of generality, we take $w_{-1} = w_0 = \prod_{k=1}^{K-1} \frac{\left(1+\frac{\alpha}{\beta}\right) \sum_{j=k+1}^K \eta_j}{\eta_k + \left(1+\frac{\alpha}{\beta}\right) \sum_{j=k+1}^K \eta_j}$, and thus $w_{K-1} = 1$. Hence, dividing both sides of Eq. (14) by $T\eta_K w_{K-1}$ and choosing the constant step size $\eta_k \equiv \eta$ for all $k \in [K]$, we obtain

$$f(\boldsymbol{x}_K) - f(\boldsymbol{x}_*) \leq \eta^2 T^2 \sigma_*^2 L \sum_{k=1}^K w_{k-1} + \frac{w_{-1}}{2\eta T} \|\boldsymbol{x}_0 - \boldsymbol{x}_*\|^2. \tag{17}$$

We first bound $w_{-1} = \prod_{k=1}^{K-1} \frac{(1+\frac{\alpha}{\beta})(K-k)}{1+(1+\frac{\alpha}{\beta})(K-k)} = \prod_{k=1}^{K-1} \frac{(1+\frac{\alpha}{\beta})k}{1+(1+\frac{\alpha}{\beta})k}$ with the constant step size. Taking the natural logarithm of $w_{-1}$, we have

$$\sum_{k=1}^{K-1} \log\left(1 - \frac{1}{1+(1+\frac{\alpha}{\beta})k}\right) \overset{(i)}{\leq} - \sum_{k=1}^{K-1} \frac{1}{1+(1+\frac{\alpha}{\beta})k} \leq -\frac{1}{1+\frac{\alpha}{\beta}} \sum_{k=1}^{K-1} \frac{1}{k+1}.$$

where for $(i)$ we use the fact that $\log(1+x) \leq x$ for $x > -1$. Further noticing that $\sum_{k=1}^{K-1} \frac{1}{k+1} = \sum_{k=1}^K \frac{1}{k} - 1 \geq \log(K) + \frac{1}{K} - 1$, then we have

$$\log(w_{-1}) \leq -\frac{1}{1+\frac{\alpha}{\beta}} \log(K) + \frac{1}{1+\frac{\alpha}{\beta}} \iff w_{-1} \leq \frac{e^{1/(1+\alpha/\beta)}}{K^{\frac{1}{1+\alpha/\beta}}} \leq \frac{e}{K^{\frac{1}{1+\alpha/\beta}}}.$$

On the other hand, we note that $w_{k-1} = \prod_{j=k}^{K-1} \frac{(1+\frac{\alpha}{\beta})(K-j)}{1+(1+\frac{\alpha}{\beta})(K-j)} = \prod_{j=1}^{K-k} \frac{(1+\frac{\alpha}{\beta})j}{1+(1+\frac{\alpha}{\beta})j}$ for $1 \leq k \leq K-1$ and $w_{K-1} = 1$, then we follow the above argument and obtain

$$\sum_{k=1}^K w_{k-1} = 1 + \sum_{k=1}^{K-1} \prod_{j=1}^{K-k} \frac{(1+\frac{\alpha}{\beta})j}{1+(1+\frac{\alpha}{\beta})j}$$

$$\leq 1 + \sum_{k=1}^{K-1} \frac{e}{(K-k+1)^{\frac{1}{1+\alpha/\beta}}} \overset{(i)}{\leq} \frac{e(1+\alpha/\beta)}{\alpha/\beta} K^{\frac{\alpha/\beta}{1+\alpha/\beta}},$$

where $(i)$ is due to $\sum_{k=2}^K \frac{1}{k^q} \leq \int_2^{K+1} \frac{1}{(x-1)^q} dx = \frac{K^{1-q}-1}{1-q}$ for any $0 < q < 1$. Hence, we obtain the final bound:

$$f(\boldsymbol{x}_K) - f(\boldsymbol{x}_*) \leq e\eta^2 T^2 \sigma_*^2 L(1+\beta/\alpha) K^{\frac{\alpha/\beta}{1+\alpha/\beta}} + \frac{e\|\boldsymbol{x}_0 - \boldsymbol{x}_*\|^2}{2T\eta K^{\frac{1}{1+\alpha/\beta}}}.$$

To analyze the oracle complexity, we take

$$\eta = \min\left\{\frac{\|\boldsymbol{x}_0 - \boldsymbol{x}_*\|^{2/3}}{2^{1/3} T \sigma_*^{2/3} L^{1/3} K^{1/3} (1+\beta/\alpha)^{1/3}}, \frac{1}{\sqrt{\beta} T L}\right\}$$

and analyze the two possible cases depending on which term in the min is smaller. If the first term in the min is smaller (which we can equivalently think of as $K$ being "large"), we get

$$f(\boldsymbol{x}_K) - f(\boldsymbol{x}_*) \leq \frac{3.5(1+\beta/\alpha)^{1/3} L^{1/3} \sigma_*^{2/3} \|\boldsymbol{x}_0 - \boldsymbol{x}_*\|^{4/3}}{K^{\frac{1}{1+\alpha/\beta} - \frac{1}{3}}}.$$

Alternatively, if $\eta = \frac{1}{\sqrt{\beta} T L} \leq \frac{\|\boldsymbol{x}_0 - \boldsymbol{x}_*\|^{2/3}}{2^{1/3} T \sigma_*^{2/3} L^{1/3} K^{1/3} (1+\beta/\alpha)^{1/3}}$ (which we can think of as having "small" $K$), we obtain

$$f(\boldsymbol{x}_K) - f(\boldsymbol{x}_*) \leq \frac{1.8(1+\beta/\alpha)^{1/3} L^{1/3} \sigma_*^{2/3} \|\boldsymbol{x}_0 - \boldsymbol{x}_*\|^{4/3}}{K^{\frac{1}{1+\alpha/\beta} - \frac{1}{3}}} + \frac{1.4\sqrt{\beta} L \|\boldsymbol{x}_0 - \boldsymbol{x}_*\|^2}{K^{\frac{1}{1+\alpha/\beta}}}.$$

Hence, combining these two cases, we have

$$f(\boldsymbol{x}_K) - f(\boldsymbol{x}_*) \leq \frac{3.5(1 + \beta/\alpha)^{1/3} L^{1/3} \sigma_*^{2/3} \|\boldsymbol{x}_0 - \boldsymbol{x}_*\|^{4/3}}{K^{\frac{1}{1+\alpha/\beta} - \frac{1}{3}}} + \frac{1.4\sqrt{\beta} L \|\boldsymbol{x}_0 - \boldsymbol{x}_*\|^2}{K^{\frac{1}{1+\alpha/\beta}}}.$$

In particular, if we choose $\alpha = 4$ and $\beta = 4 \log K$, assuming without loss of generality that $\log K > 1$, then we have $K^{\frac{\alpha/\beta}{1+\alpha/\beta}} = K^{\frac{1}{\log K + 1}} \leq e$ and thus

$$f(\boldsymbol{x}_K) - f(\boldsymbol{x}_*) \leq \frac{9.4 L^{\frac{1}{3}} \sigma_*^{\frac{2}{3}} \|\boldsymbol{x}_0 - \boldsymbol{x}_*\|^{\frac{4}{3}} (1 + \log K)^{\frac{1}{3}}}{K^{\frac{2}{3}}} + \frac{7.4 L \|\boldsymbol{x}_0 - \boldsymbol{x}_*\|^2 \sqrt{\log K}}{K}.$$

To guarantee $f(\boldsymbol{x}_K) - f(\boldsymbol{x}_*) \leq \epsilon$ given $\epsilon > 0$, the total number of individual gradient evaluations will be

$$TK = \widetilde{\mathcal{O}}\Big(\frac{TL\|\boldsymbol{x}_0 - \boldsymbol{x}_*\|^2}{\epsilon} + \frac{TL^{1/2}\sigma_*\|\boldsymbol{x}_0 - \boldsymbol{x}_*\|^2}{\epsilon^{3/2}}\Big).$$

$\square$

**Corollary 1** (Increasing Weighted Averaging). *Under Assumptions 1, 3, and 4 and for parameters $\alpha, \beta > 0$ such that $\frac{1}{\alpha} + \frac{1}{\beta} \leq \frac{1}{2}$, if the step size $\eta$ is fixed and such that $\eta \leq \frac{1}{\sqrt{\beta}TL}$, then for any constant $c \in (0,1]$ and increasing sequence $\{w_k\}_{k=0}^{K-1}$ with $w_k = \frac{(1+\alpha/\beta)(K-k)+1-c}{(1+\alpha/\beta)(K-k)} w_{k-1}$, Alg. 1 outputs $\hat{\boldsymbol{x}}_K = \sum_{k=1}^{K} \frac{w_{k-1}\boldsymbol{x}_k}{\sum_{k=1}^{K} w_{k-1}}$ satisfying*

$$f(\hat{\boldsymbol{x}}_K) - f(\boldsymbol{x}_*) \leq \frac{T^2 \sigma_*^2 \eta^2 L}{c} + \frac{\|\boldsymbol{x}_0 - \boldsymbol{x}_*\|^2}{2c\eta TK}.$$

*In particular, taking $\beta = \mathcal{O}(1)$ and the step size $\eta = \min\{\frac{1}{\sqrt{\beta}TL}, (\frac{\|\boldsymbol{x}_0-\boldsymbol{x}_*\|^2}{2T^3\sigma_*^2 LK})^{1/3}\}$, we have $f(\hat{\boldsymbol{x}}_K) - f(\boldsymbol{x}_*) \leq \epsilon$ after $\mathcal{O}\big(\frac{TL\|\boldsymbol{x}_0-\boldsymbol{x}_*\|^2}{c\epsilon} + \frac{TL^{1/2}\sigma_*\|\boldsymbol{x}_0-\boldsymbol{x}_*\|^2}{c^{3/2}\epsilon^{3/2}}\big)$ individual gradient evaluations.*

*Proof.* We follow the proof of Theorem 1 up to Eq. (14) with constant step size $\eta$, then we instead take $\lambda_k = w_{k-1}/w_k$ and $w_k = \frac{(1+\frac{\alpha}{\beta})(K-k)+1-c}{(1+\frac{\alpha}{\beta})(K-k)} w_{k-1}$ to obtain

$$c\eta T \sum_{k=1}^{K} w_{k-1}\big(f(\boldsymbol{x}_k) - f(\boldsymbol{x}_*)\big) \leq T^3 \eta^3 \sigma_*^2 L \sum_{k=1}^{K} w_{k-1} + \frac{w_{-1}}{2}\|\boldsymbol{x}_0 - \boldsymbol{x}_*\|^2.$$

Since $f$ is convex, we have $f(\hat{\boldsymbol{x}}_K) - f(\boldsymbol{x}_*) \leq \frac{\sum_{k=1}^{K} w_{k-1}(f(\boldsymbol{x}_k)-f(\boldsymbol{x}_*))}{\sum_{k=1}^{K} w_{k-1}}$ where $\hat{\boldsymbol{x}}_K = \sum_{k=1}^{K} \frac{w_{k-1}\boldsymbol{x}_k}{\sum_{k=1}^{K} w_{k-1}}$ is the increasing weighted averaging of $\{\boldsymbol{x}_k\}_{k=1}^{K}$, thus (*cf.* Eq. (17))

$$f(\hat{\boldsymbol{x}}_K) - f(\boldsymbol{x}_*) \leq \frac{T^2 \sigma_*^2 \eta^2 L}{c} + \frac{\|\boldsymbol{x}_0 - \boldsymbol{x}_*\|^2 w_{-1}}{2c\eta T \sum_{k=1}^{K} w_{k-1}} \leq \frac{T^2 \sigma_*^2 \eta^2 L}{c} + \frac{\|\boldsymbol{x}_0 - \boldsymbol{x}_*\|^2}{2c\eta TK},$$

where the last step is due to $\sum_{k=1}^{K} w_{k-1} \geq K w_{-1}$. Then we follow the proof of Theorem 1 and choose $\eta = \min\{\frac{1}{\sqrt{\beta}TL}, (\frac{\|\boldsymbol{x}_0-\boldsymbol{x}_*\|^2}{2T^3\sigma_*^2 LK})^{1/3}\}$ to obtain

$$f(\hat{\boldsymbol{x}}_K) - f(\boldsymbol{x}_*) \leq \frac{\sqrt{\beta} L \|\boldsymbol{x}_0 - \boldsymbol{x}_*\|^2}{2cK} + \frac{2^{1/3} L^{1/3} \sigma_*^{2/3} \|\boldsymbol{x}_0 - \boldsymbol{x}_*\|^{4/3}}{cK^{2/3}}.$$

To guarantee $f(\boldsymbol{x}_K) - f(\boldsymbol{x}_*) \leq \epsilon$ for $\epsilon > 0$, we choose $\beta = \mathcal{O}(1)$ and the total number of individual gradient evaluations will be

$$TK = \mathcal{O}\Big(\frac{TL\|\boldsymbol{x}_0 - \boldsymbol{x}_*\|^2}{c\epsilon} + \frac{TL^{1/2}\sigma_*\|\boldsymbol{x}_0 - \boldsymbol{x}_*\|^2}{c^{3/2}\epsilon^{3/2}}\Big),$$

thus finishing the proof. $\square$

**Corollary 2** (Shuffled SGD (RR/SO)). *Under Assumptions 1, 3 and 4 and for positive parameters $\alpha, \beta > 0$ such that $\frac{1}{\alpha} + \frac{1}{\beta} \leq \frac{1}{2}$, if the step size is fixed and such that $\eta \leq \frac{1}{\sqrt{\beta}TL}$, the output $\boldsymbol{x}_K$ of Alg. 1 with uniformly random (SO/RR) shuffling satisfies*

$$\mathbb{E}[f(\boldsymbol{x}_K) - f(\boldsymbol{x}_*)] \leq e\eta^2\sigma_*^2TL(1 + \beta/\alpha)K^{\frac{\alpha/\beta}{1+\alpha/\beta}} + \frac{e\|\boldsymbol{x}_0 - \boldsymbol{x}_*\|^2}{2T\eta K^{\frac{1}{1+\alpha/\beta}}}.$$

*With $\alpha = 4, \beta = 4\log K$ and $\eta = \min\left\{\frac{\|\boldsymbol{x}_0 - \boldsymbol{x}_*\|^{2/3}}{2^{1/3}T^{2/3}\sigma_*^{2/3}L^{1/3}K^{1/3}(1+\log K)^{1/3}}, \frac{1}{2\sqrt{\log K}TL}\right\}$, we have $\mathbb{E}[f(\boldsymbol{x}_K) - f(\boldsymbol{x}_*)] \leq \epsilon$ after $\widetilde{\mathcal{O}}\left(\frac{TL\|\boldsymbol{x}_0 - \boldsymbol{x}_*\|^2}{\epsilon} + \frac{\sqrt{TL}\sigma_*\|\boldsymbol{x}_0 - \boldsymbol{x}_*\|^2}{\epsilon^{3/2}}\right)$ individual gradient evaluations.*

Before proceeding the proof of Corollary 2, it is helpful to first introduce a lemma to bound the term $\sum_{t=1}^{T}\|\sum_{s=t}^{T}\nabla f_s(\boldsymbol{x}_*)\|^2$ in Lemma 1.

**Lemma 3.** *Under Assumptions 4 and for Alg. 1 with uniformly random (SO/RR) shuffling, we have*

$$\mathbb{E}\left[\sum_{t=1}^{T}\|\sum_{s=t}^{T}\nabla f_s(\boldsymbol{x}_*)\|^2\right] \leq \frac{T(T+1)}{6}\sigma_*^2.$$

*Proof.* We first consider the case of random reshuffling strategy. Conditional on all the randomness up to but not including $k$-th epoch, the only randomness of $\mathbb{E}_k[\sum_{t=1}^{T}\|\sum_{s=t}^{T}\nabla f_s(\boldsymbol{x}_*)\|^2]$ comes from the permutation $\pi^{(k)}$ at $k$-th epoch. Further noticing that each partial sum $\sum_{s=t}^{T}\nabla f_s(\boldsymbol{x}_*)$ can be seen as a batch sampled without replacement from $\{\nabla f_t(\boldsymbol{x}_*)\}_{t\in[T]}$, we have

$$\mathbb{E}_k\left[\sum_{t=1}^{T}\left\|\sum_{s=t}^{T}\nabla f_s(\boldsymbol{x}_*)\right\|^2\right] = \sum_{t=1}^{T}\mathbb{E}_{\pi^{(k)}}\left[\left\|\sum_{s=t}^{T}\nabla f_s(\boldsymbol{x}_*)\right\|^2\right]$$

$$= \sum_{t=1}^{T}(T-t+1)^2\mathbb{E}_{\pi^{(k)}}\left[\left\|\frac{1}{T-t+1}\sum_{s=t}^{T}\nabla f_s(\boldsymbol{x}_*)\right\|^2\right]$$

$$\overset{(i)}{\leq} \sum_{t=1}^{T}(T-t+1)^2\frac{t-1}{(T-t+1)(T-1)}\sigma_*^2$$

$$= \frac{T(T+1)}{6}\sigma_*^2,$$

where $(i)$ is due to $\sum_{t=1}^{T}\nabla f_t(\boldsymbol{x}_*) = \boldsymbol{0}$ and sampling without replacement (see e.g., (Lohr, 2021, Section 2.7)). It remains to take expectation w.r.t. all randomness on both sides and use the law of total expectation. For the case of shuffle-once variant, we can directly take expectation since the randomness only comes from the initial random permutation, and the above argument still applies. □

*Proof of Corollary 2.* This follows the analysis in the proof of Theorem 1, with taking expectation w.r.t. all the randomness on both sides of the inequality in Lemma 1 and using Lemma 3. □

## C OMITTED PROOFS FROM SECTION 3

### C.1 CONVEX SMOOTH SETTING

**Lemma 4.** *Under Assumptions 1 and 3, for any $\boldsymbol{z} \in \mathbb{R}^d$ that is fixed in the $k$-th cycle of Alg. 2 and for $\alpha > 0, \beta > 0$ such that $\frac{1}{\alpha} + \frac{1}{\beta} \leq \frac{1}{2}$, if the step sizes satisfy $\eta_k \leq \frac{1}{\sqrt{\beta}TL}$, then we have for $k \in [K]$*

$$T(f(\boldsymbol{x}_k) - f(\boldsymbol{z})) \leq \eta_k^2 L\sum_{t=1}^{T-1}\left\|\sum_{s=t+1}^{T}\nabla f_s(\boldsymbol{x}_*)\right\|^2 + \frac{\alpha}{\beta}T(f(\boldsymbol{z}) - f(\boldsymbol{x}_*))$$

$$+ \frac{1}{2\eta_k}\left(\|\boldsymbol{x}_{k-1} - \boldsymbol{z}\|^2 - \|\boldsymbol{x}_k - \boldsymbol{z}\|^2\right).$$

(18)

*Proof.* Since each $f_t$ is convex and $L$-smooth, we have for $t \in [T]$

$$f_t(\boldsymbol{x}_k) - f_t(\boldsymbol{x}_{k-1,t+1}) \leq \langle \nabla f_t(\boldsymbol{x}_k), \boldsymbol{x}_k - \boldsymbol{x}_{k-1,t+1} \rangle - \frac{1}{2L} \|\nabla f_t(\boldsymbol{x}_k) - \nabla f_t(\boldsymbol{x}_{k-1,t+1})\|^2,$$

$$f_t(\boldsymbol{x}_{k-1,t+1}) - f_t(\boldsymbol{z}) \leq \langle \nabla f_t(\boldsymbol{x}_{k-1,t+1}), \boldsymbol{x}_{k-1,t+1} - \boldsymbol{z} \rangle - \frac{1}{2L} \|\nabla f_t(\boldsymbol{x}_{k-1,t+1}) - \nabla f_t(\boldsymbol{z})\|^2.$$

Following the proof of Lemma 1, we add and subtract $\frac{1}{2\eta_k}\|\boldsymbol{x}_{k-1,t} - \boldsymbol{z}\|^2$ on the right-hand side of the second inequality and combine the above two inequalities to obtain

$$
\begin{aligned}
f_t(\boldsymbol{x}_k) - f_t(\boldsymbol{z}) \leq\ & \langle \nabla f_t(\boldsymbol{x}_k), \boldsymbol{x}_k - \boldsymbol{x}_{k-1,t+1} \rangle - \frac{1}{2\eta_k} \|\boldsymbol{x}_{k-1,t+1} - \boldsymbol{x}_{k-1,t}\|^2 \\
& - \frac{1}{2L} \left( \|\nabla f_t(\boldsymbol{x}_k) - \nabla f_t(\boldsymbol{x}_{k-1,t+1})\|^2 + \|\nabla f_t(\boldsymbol{x}_{k-1,t+1}) - \nabla f_t(\boldsymbol{z})\|^2 \right) \\
& + \frac{1}{2\eta_k} \left( \|\boldsymbol{x}_{k-1,t} - \boldsymbol{z}\|^2 - \|\boldsymbol{x}_{k-1,t+1} - \boldsymbol{z}\|^2 \right).
\end{aligned}
$$

Decomposing $\nabla f_t(\boldsymbol{x}_k) = \nabla f_t(\boldsymbol{x}_k) - \nabla f_t(\boldsymbol{x}_{k-1,t+1}) + \nabla f_t(\boldsymbol{x}_{k-1,t+1})$ and summing over $t \in [T]$, we note that $\boldsymbol{x}_{k-1} = \boldsymbol{x}_{k-1,1}$ and $\boldsymbol{x}_k = \boldsymbol{x}_{k-1,T+1}$ and obtain

$$
\begin{aligned}
& T\big(f(\boldsymbol{x}_k) - f(\boldsymbol{z})\big) \\
\leq\ & \underbrace{\sum_{t=1}^{T} \langle \nabla f_t(\boldsymbol{x}_{k-1,t+1}), \boldsymbol{x}_k - \boldsymbol{x}_{k-1,t+1} \rangle - \frac{1}{2\eta_k} \sum_{t=1}^{T} \|\boldsymbol{x}_{k-1,t+1} - \boldsymbol{x}_{k-1,t}\|^2}_{\mathcal{T}_1} \\
& + \underbrace{\sum_{t=1}^{T} \langle \nabla f_t(\boldsymbol{x}_k) - \nabla f_t(\boldsymbol{x}_{k-1,t+1}), \boldsymbol{x}_k - \boldsymbol{x}_{k-1,t+1} \rangle}_{\mathcal{T}_2} + \frac{1}{2\eta_k} \left( \|\boldsymbol{x}_{k-1} - \boldsymbol{z}\|^2 - \|\boldsymbol{x}_k - \boldsymbol{z}\|^2 \right) \\
& - \frac{1}{2L} \sum_{t=1}^{T} \|\nabla f_t(\boldsymbol{x}_k) - \nabla f_t(\boldsymbol{x}_{k-1,t+1})\|^2 - \frac{1}{2L} \sum_{t=1}^{T} \|\nabla f_t(\boldsymbol{x}_{k-1,t+1}) - \nabla f_t(\boldsymbol{z})\|^2.
\end{aligned}
$$

For the term $\mathcal{T}_1$, we follow the argument from the proof of Theorem 1 to obtain

$$\mathcal{T}_1 = -\frac{1}{2\eta_k} \|\boldsymbol{x}_k - \boldsymbol{x}_{k-1}\|^2 \leq 0.$$

For the term $\mathcal{T}_2$, noticing that $\boldsymbol{x}_k - \boldsymbol{x}_{k-1,t+1} = -\eta_k \sum_{s=t+1}^{T} \nabla f_s(\boldsymbol{x}_{k-1,s+1})$ for $1 \leq t \leq T-1$ and decomposing $\nabla f_s(\boldsymbol{x}_{k-1,s+1}) = (\nabla f_s(\boldsymbol{x}_{k-1,s+1}) - \nabla f_s(\boldsymbol{z})) + (\nabla f_s(\boldsymbol{z}) - \nabla f_s(\boldsymbol{x}_*)) + \nabla f_s(\boldsymbol{x}_*)$, we use Young's inequality with parameters $\alpha > 0$ and $\beta > 0$ to obtain

$$
\begin{aligned}
\mathcal{T}_2 =\ & \sum_{t=1}^{T-1} \left\langle \nabla f_t(\boldsymbol{x}_k) - \nabla f_t(\boldsymbol{x}_{k-1,t+1}), -\eta_k \sum_{s=t+1}^{T} \nabla f_s(\boldsymbol{x}_{k-1,s+1}) \right\rangle \\
\leq\ & \frac{1}{2L} \left( \frac{1}{2} + \frac{1}{\alpha} + \frac{1}{\beta} \right) \sum_{t=1}^{T} \|\nabla f_t(\boldsymbol{x}_k) - \nabla f_t(\boldsymbol{x}_{k-1,t+1})\|^2 \\
& + \frac{\alpha \eta_k^2 L}{2} \sum_{t=1}^{T-1} \left\| \sum_{s=t+1}^{T} (\nabla f_s(\boldsymbol{z}) - \nabla f_s(\boldsymbol{x}_*)) \right\|^2 + \eta_k^2 L \sum_{t=1}^{T-1} \left\| \sum_{s=t+1}^{T} \nabla f_s(\boldsymbol{x}_*) \right\|^2 \\
& + \frac{\beta \eta_k^2 L}{2} \sum_{t=1}^{T-1} \left\| \sum_{s=t+1}^{T} (\nabla f_s(\boldsymbol{x}_{k-1,s+1}) - \nabla f_s(\boldsymbol{z})) \right\|^2.
\end{aligned}
$$

Further using the fact that $\|\sum_{i=1}^n \boldsymbol{x}_i\|^2 \leq n \sum_{i=1}^n \|\boldsymbol{x}_i\|^2$ and combining the above bounds on $\mathcal{T}_1$ and $\mathcal{T}_2$, we obtain

$$
\begin{aligned}
T\big(f(\boldsymbol{x}_k) - f(\boldsymbol{z})\big) \leq{} & \frac{1}{2L}\Big(\frac{1}{\alpha} + \frac{1}{\beta} - \frac{1}{2}\Big) \sum_{t=1}^T \|\nabla f_t(\boldsymbol{x}_k) - \nabla f_t(\boldsymbol{x}_{k-1,t})\|^2 \\
& + \Big(\frac{\beta \eta_k^2 T^2 L}{2} - \frac{1}{2L}\Big) \sum_{t=1}^T \|\nabla f_t(\boldsymbol{x}_{k-1,t+1}) - \nabla f_t(\boldsymbol{z})\|^2 \\
& + \frac{\alpha \eta_k^2 T^2 L}{2} \sum_{t=1}^T \|\nabla f_t(\boldsymbol{z}) - \nabla f_t(\boldsymbol{x}_*)\|^2 + \eta_k^2 L \sum_{t=1}^{T-1} \Big\| \sum_{s=t+1}^T \nabla f_s(\boldsymbol{x}_*)\Big\|^2 \\
& + \frac{1}{2\eta_k}\big(\|\boldsymbol{x}_{k-1} - \boldsymbol{z}\|^2 - \|\boldsymbol{x}_k - \boldsymbol{z}\|^2\big).
\end{aligned}
$$

The rest of the proof is the same as the proof of Lemma 1 and is thus omitted. $\qquad\square$

**Theorem 2.** *Under Assumptions 1, 3, and 4 and for parameters $\alpha, \beta > 0$ such that $\frac{1}{\alpha} + \frac{1}{\beta} \leq \frac{1}{2}$, if the step size is fixed and such that $\eta \leq \frac{1}{\sqrt{\beta}TL}$, the output $\boldsymbol{x}_K$ of Alg. 2 satisfies*

$$
f(\boldsymbol{x}_K) - f(\boldsymbol{x}_*) \leq \mathrm{e}\eta^2 T^2 \sigma_*^2 L(1 + \beta/\alpha) K^{\frac{\alpha/\beta}{1+\alpha/\beta}} + \frac{\mathrm{e}\|\boldsymbol{x}_0 - \boldsymbol{x}_*\|^2}{2\eta T K^{\frac{1}{1+\alpha/\beta}}}.
$$

*With $\alpha = 4, \beta = 4\log K$ and $\eta = \min\Big\{\frac{\|\boldsymbol{x}_0 - \boldsymbol{x}_*\|^{2/3}}{2^{1/3}T\sigma_*^{2/3}L^{1/3}K^{1/3}(1+\log K)^{1/3}}, \frac{1}{2\sqrt{\log K}TL}\Big\}$, we have $f(\boldsymbol{x}_K) - f(\boldsymbol{x}_*) = \mathcal{O}\Big(\frac{L^{1/3}\sigma_*^{2/3}\|\boldsymbol{x}_0 - \boldsymbol{x}_*\|^{4/3}(1+\log K)^{1/3}}{K^{2/3}} + \frac{L\|\boldsymbol{x}_0 - \boldsymbol{x}_*\|^2\sqrt{\log K}}{K}\Big)$, and thus $f(\boldsymbol{x}_K) - f(\boldsymbol{x}_*) \leq \epsilon$ after $\widetilde{\mathcal{O}}\Big(\frac{TL\|\boldsymbol{x}_0 - \boldsymbol{x}_*\|^2}{\epsilon} + \frac{TL^{1/2}\sigma_*\|\boldsymbol{x}_0 - \boldsymbol{x}_*\|^2}{\epsilon^{3/2}}\Big)$ individual gradient evaluations.*

*Proof.* This follows Lemma 4 and the proof of Theorem 1. $\qquad\square$

**Theorem 3.** *Given $L > 0$ and $T \geq 2$, let $\mathcal{F}_{T,L}$ be the class of finite-sum functions $f(\boldsymbol{x}) = \frac{1}{T}\sum_{t=1}^T f_t(\boldsymbol{x})$, with each component $f_t$ being $L$-smooth and convex. Then:*

*1. For any fixed step size $\eta$, there exists a function $f \in \mathcal{F}_{T,L}$ such that the iterates $\boldsymbol{x}_k$ of Alg. 2 satisfy $f(\boldsymbol{x}_k) - f(\boldsymbol{x}_*) \nrightarrow 0$ as $k \to \infty$. As a consequence, the forgetting is catastrophic.*

*2. For any fixed step size $\eta$ that only depends on the parameters of the problem class ($L, T$, error $\epsilon$), there exists a function $f \in \mathcal{F}_{T,L}$ such that the iterates $\boldsymbol{x}_k$ of Alg. 2 satisfy $\lim_{k\to\infty} f(\boldsymbol{x}_k) - f(\boldsymbol{x}_*) > 1$.*

*3. Given $\varepsilon > 0$, if the fixed step size $\eta$ satisfies $\eta \geq \min\Big\{\frac{16\sqrt{\varepsilon}}{\sqrt{T}L\sigma_*}, \frac{1}{TL}\Big\}$, then there exists a function $f \in \mathcal{F}_{T,L}$ such that the iterates $\boldsymbol{x}_k$ of Alg. 2 satisfy $f(\boldsymbol{x}_k) - f(\boldsymbol{x}_*) > \varepsilon$ for all sufficiently large $k$.*

*Proof.* For all parts of the proof, we consider 1-dimensional quadratics

$$
f(x) := \frac{1}{T}\sum_{t=1}^T f_t(x), \quad \text{where} \quad f_t(x) = \frac{L}{2}(x - \delta_t)^2
$$

for $t \in [T]$, $L > 0$, and appropriately chosen sequences of $\{\delta_t\}_{t\in[T]} \subseteq \mathbb{R}$.

It is immediate that $f(x)$ is minimized at $x_* = \frac{1}{T}\sum_{t=1}^T \delta_t$. Observe that Alg. 2 using a constant step size $\eta > 0$ has closed-form updates on $f$, i.e.,

$$
x_{k+1} = \gamma^n x_k + (1 - \gamma)\sum_{t=1}^T \gamma^{T-t}\delta_t,
$$

where $\gamma = \frac{1}{\eta L + 1} \in (0, 1)$. Given any initial point $x_0$, by iterating we have

$$x_k - x_* = \gamma^{kT} x_0 + \sum_{t=1}^{T} \left( \frac{\gamma^{T-t}(1-\gamma)(1-\gamma^{kT})}{1-\gamma^T} - \frac{1}{T} \right) \delta_t \tag{19}$$

$$\xrightarrow{k \to \infty} \sum_{t=1}^{T} \left( \frac{\gamma^{T-t}(1-\gamma)}{1-\gamma^T} - \frac{1}{T} \right) \delta_t. \tag{20}$$

1. Consider the weight $\delta_T$ in Eq. (20). Since $\gamma \in (0, 1)$, we have

$$\frac{1-\gamma}{1-\gamma^T} - \frac{1}{T} = \frac{1}{\sum_{t=0}^{T-1} \gamma^t} - \frac{1}{T} > 0.$$

Then for any $\{\delta_t\}_{t \in [T]}$ such that $\mathrm{sgn}(\delta_t) = \mathrm{sgn}\left( \frac{\gamma^{T-t}(1-\gamma)}{1-\gamma^T} - \frac{1}{T} \right)$ and $\delta_T > 0$, we know that

$$\lim_{k \to \infty} f(x_k) - f(x_*) \stackrel{(i)}{=} \frac{L}{2} \lim_{k \to \infty} (x_k - x_*)^2 \geq \frac{L}{2} \left( \frac{1-\gamma}{1-\gamma^T} - \frac{1}{T} \right)^2 \delta_T^2 > 0,$$

where $(i)$ is due to $f$ being both $L$-strongly convex and $L$-smooth.

2. Consider the weights of $\delta_t$ in Eq. (20). Since $\gamma \in (0, 1)$, we have

$$0 \leq \frac{\gamma^{T-t}(1-\gamma)}{1-\gamma^T} \leq \gamma^{T-t} \leq 1,$$

thus for any $t \in [T-1]$

$$\frac{\gamma^{T-t}(1-\gamma)}{1-\gamma^T} - \frac{1}{T} \in \left[ -\frac{1}{T}, \frac{T-1}{T} \right).$$

For $t = T$, given any fixed $\gamma < 1$, we have

$$\frac{1-\gamma}{1-\gamma^T} - \frac{1}{T} = \frac{1}{\sum_{t=0}^{T-1} \gamma^t} - \frac{1}{T} > 0.$$

Hence, for the sequence $\{\delta_t\}$ such that

$$|\delta_t| < \frac{T\sqrt{2/L}}{(T-1)^2} \ (t \in [T-1]), \quad \delta_T > \frac{2\sqrt{2/L}}{\frac{1-\gamma}{1-\gamma^T} - \frac{1}{T}},$$

then combining the bounds on the weights of $\delta_t$ with Eq. (20) we obtain

$$\lim_{k \to \infty} x_k - x_* = \sum_{t=1}^{T} \left( \frac{\gamma^{T-t}(1-\gamma)}{1-\gamma^T} - \frac{1}{T} \right) \delta_t$$

$$\geq \left( \frac{1-\gamma}{1-\gamma^T} - \frac{1}{T} \right) \delta_T - \sum_{t=1}^{T-1} \frac{T-1}{T} |\delta_t|$$

$$> \sqrt{2/L}.$$

Since $f$ is $L$-smooth and $L$-strongly convex, we know that

$$\lim_{k \to \infty} f(x_k) - f(x_*) = \frac{L}{2} \lim_{k \to \infty} (x_k - x_*)^2 > 1,$$

thus finishing the proof of the second part. We note in passing that 1 on the right-hand side can be replaced by any constant using a simple rescaling.

3. Observe that given a fixed step size $\eta > 0$, we can choose a sequence $\{\delta_t\}_{t\in[T]}$ such that $\mathrm{sgn}(\delta_t) = \mathrm{sgn}\big(\frac{\gamma^{T-t}(1-\gamma)}{1-\gamma^T} - \frac{1}{T}\big)$ for all $t \in [T]$, thus for any initial point $x_0 \geq x_*$:

$$f(x_k) - f(x_*) = \frac{L}{2}\Big(\gamma^{kT}(x_0 - x_*) + (1 - \gamma^{kT})\sum_{t=1}^{T}\Big(\frac{\gamma^{T-t}(1-\gamma)}{1-\gamma^T} - \frac{1}{T}\Big)\delta_t\Big)^2$$

$$\geq \frac{(1 - \gamma^{kT})^2 L}{2}\sum_{t=1}^{T}\Big(\frac{\gamma^{T-t}(1-\gamma)}{1-\gamma^T} - \frac{1}{T}\Big)^2\delta_t^2$$

$$+ (1 - \gamma^{kT})^2 L \sum_{s\neq t\in[T]}\Big(\frac{\gamma^{T-t}(1-\gamma)}{1-\gamma^T} - \frac{1}{T}\Big)\Big(\frac{\gamma^{T-t}(1-\gamma)}{1-\gamma^T} - \frac{1}{T}\Big)\delta_s\delta_t.$$

Without loss of generality, taking a sufficiently large $k \geq \frac{\log_\gamma(1-2/\sqrt{5})}{T}$, we obtain

$$f(x_k) - f(x_*) \geq \frac{2L}{5}\sum_{t=1}^{T}\Big(\frac{\gamma^{T-t}(1-\gamma)}{1-\gamma^T} - \frac{1}{T}\Big)^2\delta_t^2.$$

Then for any step size $\eta \geq 1/TL$, we have $\gamma = \frac{1}{\eta L+1} \leq \frac{T}{T+1}$. Consider $t = T$, we can bound

$$\frac{1-\gamma}{1-\gamma^T} - \frac{1}{T} \geq \frac{1 - \frac{T}{T+1}}{1 - (\frac{T}{T+1})^T} - \frac{1}{T} = \frac{1}{T}\Big(\frac{1}{1 - (1 - \frac{1}{T})^T} - 1\Big) > \frac{e-1}{T}.$$

Thus, for $\delta_T \geq \frac{\sqrt{5}T\sqrt{\varepsilon}}{\sqrt{2}(e-1)L}$, we have

$$f(x_k) - f(x_*) \geq \frac{2L}{5}\Big(\frac{1-\gamma}{1-\gamma^T} - \frac{1}{T}\Big)^2\delta_T^2 > \varepsilon.$$

On the other hand, recalling the definition in Assumption 4, we have in this example that

$$\sigma_*^2 = \frac{L^2}{T}\sum_{t=1}^{T}\Big(\frac{\sum_{t=1}^{T}\delta_t}{T} - \delta_t\Big)^2 = \frac{L^2}{T}\Big(\frac{(\sum_{t=1}^{T}\delta_t)^2}{T} - 2\frac{(\sum_{t=1}^{T}\delta_t)^2}{T} + \sum_{t=1}^{T}\delta_t^2\Big) \leq L^2\sum_{t=1}^{T}\delta_t^2/T.$$

Thus for any step size $\eta \geq \frac{16\sqrt{\varepsilon}}{\sqrt{T}L\sigma_*} \geq \frac{16\sqrt{\varepsilon}}{L^{3/2}\sqrt{\sum_{t=1}^{T}\delta_t^2}}$, we have $\gamma = \frac{1}{\eta L+1} \leq \frac{\sqrt{\sum_{t=1}^{T}\delta_t^2}}{16\sqrt{\varepsilon/L}+\sqrt{\sum_{t=1}^{T}\delta_t^2}}$.

We now proceed by bounding the weight of $\delta_T$. In particular, let $\gamma = 1 - \kappa$ for some $\kappa > 0$, and assume that $\kappa \leq \frac{1}{T+1} < \frac{1}{T}$ without loss of generality by the discussion above. Since $\kappa T < 1$ and $T \geq 2$, we have

$$(1 - \kappa)^T \geq 1 - \kappa T + \frac{\kappa^2 T(T-1)}{4},$$

which leads to

$$1 - \gamma^T = 1 - (1 - \kappa)^T \leq \kappa T - \frac{\kappa^2 T(T-1)}{4}.$$

Hence, we have

$$\frac{1-\gamma}{1-\gamma^T} \geq \frac{\kappa}{\kappa T - \frac{\kappa^2 T(T-1)}{4}} = \frac{1}{T}\frac{1}{1 - \kappa(T-1)/4}.$$

Further noticing that

$$\frac{1}{1 - \kappa(T-1)/4} \geq 1 + \frac{\kappa(T-1)}{4} \geq 1 + \frac{\kappa T}{8}$$

for $T \geq 2$ and $\kappa < \frac{1}{T}$, then we have

$$\frac{1-\gamma}{1-\gamma^T} - \frac{1}{T} \geq \frac{1}{T}\Big(1 + \frac{\kappa T}{8}\Big) - \frac{1}{T} = \frac{\kappa}{8}.$$

Recall that $\gamma \leq \frac{\sqrt{\sum_{t=1}^{T}\delta_t^2}}{16\sqrt{\varepsilon/L}+\sqrt{\sum_{t=1}^{T}\delta_t^2}}$, then we obtain

$$\frac{1-\gamma}{1-\gamma^T}-\frac{1}{T} \geq \frac{2\sqrt{\varepsilon/L}}{16\sqrt{\varepsilon/L}+\sqrt{\sum_{t=1}^{T}\delta_t^2}} \geq \frac{\sqrt{3\varepsilon/L}}{\sqrt{\sum_{t=1}^{T}\delta_t^2}},$$

for $\{\delta_t\}_{t\in[T]}$ such that $\sqrt{\sum_{t=1}^{T}\delta_t^2} \geq 16\sqrt{3}(2+\sqrt{3})\sqrt{\varepsilon/L}$. Thus, for the sequence $\{\delta_t\}_{t\in[T]}$ such that $\delta_T^2 > \frac{5}{6}\sum_{t=1}^{T}\delta_t^2$ and $\mathrm{sgn}(\delta_t) = \mathrm{sgn}\left(\frac{\gamma^{T-t}(1-\gamma)}{1-\gamma^T}-\frac{1}{T}\right)$, we have

$$f(x_k) - f(x_*) \geq \frac{2L}{5}\left(\frac{(1-\gamma)}{1-\gamma^T}-\frac{1}{T}\right)^2\delta_T^2$$

$$> \frac{2L}{5}\frac{3\varepsilon\delta_T^2}{L\sum_{t=1}^{T}\delta_t^2}$$

$$> \varepsilon,$$

completing the proof. □

## C.2 CONVEX LIPSCHITZ SETTING

**Lemma 5.** *Under Assumptions 1 and 2, for any $z \in \mathbb{R}^d$ that is fixed in the $k$-th cycle of Alg. 2, we have for $k \in [K]$*

$$T\big(f(\boldsymbol{x}_k) - f(\boldsymbol{z})\big) \leq \frac{1}{2\eta_k}\big(\|\boldsymbol{x}_{k-1}-\boldsymbol{z}\|^2 - \|\boldsymbol{x}_k-\boldsymbol{z}\|^2\big) + \frac{T(T-1)G^2\eta_k}{2}. \quad (21)$$

*Proof.* Since $f_t$ is convex and closed, we have

$$\nabla M_{\eta_k f_t}(\boldsymbol{x}_{k-1,t}) = \frac{1}{\eta_k}(\boldsymbol{x}_{k-1,t}-\boldsymbol{x}_{k-1,t+1}) \in \partial f_t(\boldsymbol{x}_{k-1,t+1}).$$

By $G$-Lipschitzness of each component function, we have that for $t \in [T-1]$

$$f_t(\boldsymbol{x}_k) - f_t(\boldsymbol{x}_{k-1,t+1})$$

$$\leq G\|\boldsymbol{x}_k - \boldsymbol{x}_{k-1,t+1}\| \leq \eta_k G \sum_{s=t+1}^{T}\|\nabla M_{\eta_k f_s}(\boldsymbol{x}_{k-1,s})\| \leq (T-t)G^2\eta_k. \quad (22)$$

On the other hand, using convexity of $f_t$, we have that for $t \in [T]$

$$f_t(\boldsymbol{z}) \geq f_t(\boldsymbol{x}_{k-1,t+1}) + \langle \nabla M_{\eta_k f_t}(\boldsymbol{x}_{k-1,t}), \boldsymbol{z}-\boldsymbol{x}_{k-1,t+1}\rangle.$$

Expanding the inner product in the above inequality leads to

$$f_t(\boldsymbol{x}_{k-1,t+1}) - f_t(\boldsymbol{z})$$

$$\leq -\frac{1}{\eta_k}\langle \boldsymbol{x}_{k-1,t}-\boldsymbol{x}_{k-1,t+1}, \boldsymbol{z}-\boldsymbol{x}_{k-1,t+1}\rangle$$

$$= \frac{1}{2\eta_k}\big(\|\boldsymbol{x}_{k-1,t}-\boldsymbol{z}\|^2 - \|\boldsymbol{x}_{k-1,t+1}-\boldsymbol{z}\|^2\big) - \frac{1}{2\eta_k}\|\boldsymbol{x}_{k-1,t+1}-\boldsymbol{x}_{k-1,t}\|^2. \quad (23)$$

Combining Eq. (22) and (23) and noticing that $\boldsymbol{x}_{k-1,T+1} = \boldsymbol{x}_k$ and $\boldsymbol{x}_{k-1,1} = \boldsymbol{x}_{k-1}$, we sum the inequalities over $t \in [T]$ and obtain

$$T(f(\boldsymbol{x}_k) - f(\boldsymbol{z})) \leq \frac{1}{2\eta_k}\big(\|\boldsymbol{x}_{k-1}-\boldsymbol{z}\|^2 - \|\boldsymbol{x}_k-\boldsymbol{z}\|^2\big) + \frac{T(T-1)G^2\eta_k}{2}$$

$$-\frac{1}{2\eta_k}\sum_{t=1}^{T}\|\boldsymbol{x}_{k-1,t+1}-\boldsymbol{x}_{k-1,t}\|^2$$

$$\leq \frac{1}{2\eta_k}\big(\|\boldsymbol{x}_{k-1}-\boldsymbol{z}\|^2 - \|\boldsymbol{x}_k-\boldsymbol{z}\|^2\big) + \frac{T(T-1)G^2\eta_k}{2}.$$

□

**Theorem 4.** *Under Assumptions 1 and 2, the output $\boldsymbol{x}_K$ of Alg. 2 satisfies*

$$f(\boldsymbol{x}_K) - f(\boldsymbol{x}_*) \leq \frac{1}{2T\sum_{k=1}^{K}\eta_k}\|\boldsymbol{x}_0 - \boldsymbol{x}_*\|^2 + \frac{G^2T}{2}\sum_{k=1}^{K}\frac{\eta_k^2}{\sum_{j=k}^{K}\eta_j}.$$

*Moreover, given $\epsilon > 0$, there exists a constant step size $\eta = \frac{\|\boldsymbol{x}_0 - \boldsymbol{x}_*\|}{GT\sqrt{K}}$ such that $f(\boldsymbol{x}_K) - f(\boldsymbol{x}_*) \leq \epsilon$ after $\widetilde{\mathcal{O}}\left(\frac{G^2T\|\boldsymbol{x}_0 - \boldsymbol{x}_*\|^2}{\epsilon^2}\right)$ individual proximal oracle queries.*

*Proof.* Plugging $\boldsymbol{z}_{k-1}$ defined in Eq. (4) into Eq. (21) and multiplying $\eta_k w_{k-1}$ on both sides, we obtain

$$T\eta_k w_{k-1}\big(f(\boldsymbol{x}_k) - f(\boldsymbol{z}_{k-1})\big)$$
$$\leq \frac{1}{2}\big(w_{k-2}\|\boldsymbol{x}_{k-1} - \boldsymbol{z}_{k-2}\|^2 - w_{k-1}\|\boldsymbol{x}_k - \boldsymbol{z}_{k-1}\|^2\big) + \frac{T(T-1)G^2\eta_k^2 w_{k-1}}{2},$$

Summing over $k \in [K]$ and using the second part of Lemma 2, we have

$$\sum_{k=1}^{K} T\eta_k\Big[w_{k-1}(f(\boldsymbol{x}_k) - f(\boldsymbol{x}_*)) - \sum_{j=0}^{k-1}w_j(1-\lambda_j)(f(\boldsymbol{x}_j) - f(\boldsymbol{x}_*))\Big]$$

$$\leq \frac{T(T-1)G^2}{2}\sum_{k=1}^{K}\eta_k^2 w_{k-1} + \frac{w_{-1}}{2}\|\boldsymbol{x}_0 - \boldsymbol{x}_*\|^2,$$

where we also recall that $\boldsymbol{z}_{-1} = \boldsymbol{x}_*$. Unrolling the terms on the left-hand side as Eq. (13) and choosing $\lambda_0 = 1$, we obtain

$$T\eta_K w_{K-1}\big(f(\boldsymbol{x}_K) - f(\boldsymbol{x}_*)\big)$$
$$+ T\sum_{k=1}^{K-1}\Big[\eta_k w_{k-1} - w_k(1-\lambda_k)\sum_{j=k+1}^{K}\eta_j\Big]\big(f(\boldsymbol{x}_k) - f(\boldsymbol{x}_*)\big) \tag{24}$$
$$\leq \frac{T(T-1)G^2}{2}\sum_{k=1}^{K}\eta_k^2 w_{k-1} + \frac{w_{-1}}{2}\|\boldsymbol{x}_0 - \boldsymbol{x}_*\|^2.$$

To obtain the last iterate guarantee, we choose $\lambda_k$ and $w_k$ such that

$$\lambda_k w_k \leq w_{k-1}, \quad 0 \leq k \leq K-1,$$

$$\eta_k w_{k-1} - w_k(1-\lambda_k)\sum_{j=k+1}^{K}\eta_j \geq 0, \quad 1 \leq k \leq K-1.$$

For simplicity and without loss of generality, we make both inequalities tight and choose $w_k = \frac{\sum_{j=k}^{K}\eta_j}{\sum_{j=k+1}^{K}\eta_j}w_{k-1}$. In particular, we choose $w_k = \frac{\eta_K}{\sum_{j=k+1}^{K}\eta_j}$ for $0 \leq k \leq K-1$ such that $w_{K-1} = 1$, then we divide $T\eta_K$ on both sides of Eq. (24) and obtain

$$f(\boldsymbol{x}_K) - f(\boldsymbol{x}_*) \leq \frac{w_{-1}}{2T\eta_K}\|\boldsymbol{x}_0 - \boldsymbol{x}_*\|^2 + \frac{G^2T}{2\eta_K}\sum_{k=1}^{K}\eta_k^2 w_{k-1}$$

$$= \frac{1}{2T\sum_{k=1}^{K}\eta_k}\|\boldsymbol{x}_0 - \boldsymbol{x}_*\|^2 + \frac{G^2T}{2}\sum_{k=1}^{K}\frac{\eta_k^2}{\sum_{j=k}^{K}\eta_j}.$$

Finally, choosing $\eta_k \equiv \eta = \frac{\|\boldsymbol{x}_0 - \boldsymbol{x}_*\|}{GT\sqrt{K}}$, we get

$$f(\boldsymbol{x}_K) - f(\boldsymbol{x}_*) \leq \frac{G\|\boldsymbol{x}_0 - \boldsymbol{x}_*\|}{2\sqrt{K}}\Big(1 + \sum_{k=1}^{K}\frac{1}{K-k+1}\Big) \leq \frac{G\|\boldsymbol{x}_0 - \boldsymbol{x}_*\|(1 + \log K/2)}{\sqrt{K}}.$$

Hence, given $\epsilon > 0$, to guarantee $f(\boldsymbol{x}_K) - f(\boldsymbol{x}_*) \leq \epsilon$, the total number of individual gradient evaluations will be

$$TK = \widetilde{\mathcal{O}}\Big(\frac{G^2T\|\boldsymbol{x}_0 - \boldsymbol{x}_*\|^2}{\epsilon^2}\Big),$$

completing the proof. $\qquad\square$

### C.3 INEXACT PROXIMAL POINT EVALUATIONS

We first prove the convergence results for convex smooth settings. The following techical lemma bounds $f(\boldsymbol{x}_k) - f(\boldsymbol{z})$ within each epoch with inexact proximal point evaluations.

**Lemma 6.** *Under Assumptions 1 and 3, for any $\boldsymbol{z} \in \mathbb{R}^d$ that is fixed in the $k$-th cycle of Alg. 2 and for $\alpha > 0, \beta > 0$ such that $\frac{1}{\alpha} + \frac{1}{\beta} \leq \frac{1}{2}$, if the step sizes satisfy $\eta_k \leq \frac{1}{\sqrt{\beta}TL}$, then we have for $k \in [K]$*

$$T\big(f(\boldsymbol{x}_k) - f(\boldsymbol{z})\big) \leq 2\eta_k^2 L \sum_{t=1}^{T-1} \Big\| \sum_{s=t+1}^{T} \nabla f_s(\boldsymbol{x}_*) \Big\|^2 + \frac{\alpha}{\beta}T\big(f(\boldsymbol{z}) - f(\boldsymbol{x}_*)\big) + \frac{T}{\eta_k}\sum_{t=1}^{T}\varepsilon_{k-1,t}^2$$

$$+ \frac{1}{2\eta_k}\|\boldsymbol{x}_{k-1} - \boldsymbol{z}\|^2 - \frac{1}{2\eta_k}\Big(1 - \frac{\sum_{t=1}^{T}\varepsilon_{k-1,t}}{\sqrt{\eta_k}}\Big)\|\boldsymbol{x}_k - \boldsymbol{z}\|^2 + \frac{\sum_{t=1}^{T}\varepsilon_{k-1,t}}{2\sqrt{\eta_k}}.$$

*Proof.* Since each $f_t$ is convex and $L$-smooth, we have for $t \in [T]$

$$f_t(\boldsymbol{x}_k) - f_t(\boldsymbol{x}_{k-1,t+1}) \leq \langle \nabla f_t(\boldsymbol{x}_k), \boldsymbol{x}_k - \boldsymbol{x}_{k-1,t+1} \rangle - \frac{1}{2L}\|\nabla f_t(\boldsymbol{x}_k) - \nabla f_t(\boldsymbol{x}_{k-1,t+1})\|^2,$$

$$f_t(\boldsymbol{x}_{k-1,t+1}) - f_t(\boldsymbol{z}) \leq \langle \nabla f_t(\boldsymbol{x}_{k-1,t+1}), \boldsymbol{x}_{k-1,t+1} - \boldsymbol{z} \rangle - \frac{1}{2L}\|\nabla f_t(\boldsymbol{x}_{k-1,t+1}) - \nabla f_t(\boldsymbol{z})\|^2.$$

Following the proof of Lemma 1, we add and subtract $\frac{1}{2\eta_k}\|\boldsymbol{x}_{k-1,t} - \boldsymbol{z}\|^2$ on the right-hand side of the second inequality and notice that $\langle \boldsymbol{g}_{k-1,t}, \boldsymbol{z} \rangle + \frac{1}{2\eta_k}\|\boldsymbol{x}_{k-1,t} - \boldsymbol{z}\|^2 \geq \langle \boldsymbol{g}_{k-1,t}, \boldsymbol{x}_{k-1,t+1} \rangle + \frac{1}{2\eta_k}\|\boldsymbol{x}_{k-1,t} - \boldsymbol{x}_{k-1,t+1}\|^2 + \frac{1}{2\eta_k}\|\boldsymbol{x}_{k-1,t+1} - \boldsymbol{z}\|^2$, then we combine the above inequalities to obtain

$$f_t(\boldsymbol{x}_k) - f_t(\boldsymbol{z}) \leq \langle \nabla f_t(\boldsymbol{x}_k), \boldsymbol{x}_k - \boldsymbol{x}_{k-1,t+1} \rangle + \langle \nabla f_t(\boldsymbol{x}_{k-1,t+1}) - \boldsymbol{g}_{k-1,t}, \boldsymbol{x}_{k-1,t+1} - \boldsymbol{z} \rangle$$

$$- \frac{1}{2L}\Big(\|\nabla f_t(\boldsymbol{x}_k) - \nabla f_t(\boldsymbol{x}_{k-1,t+1})\|^2 + \|\nabla f_t(\boldsymbol{x}_{k-1,t+1}) - \nabla f_t(\boldsymbol{z})\|^2\Big)$$

$$- \frac{1}{2\eta_k}\|\boldsymbol{x}_{k-1,t+1} - \boldsymbol{x}_{k-1,t}\|^2 + \frac{1}{2\eta_k}\Big(\|\boldsymbol{x}_{k-1,t} - \boldsymbol{z}\|^2 - \|\boldsymbol{x}_{k-1,t+1} - \boldsymbol{z}\|^2\Big).$$

We decompose $\nabla f_t(\boldsymbol{x}_k) = \nabla f_t(\boldsymbol{x}_k) - \nabla f_t(\boldsymbol{x}_{k-1,t+1}) + \nabla f_t(\boldsymbol{x}_{k-1,t+1}) - \boldsymbol{g}_{k-1,t} + \boldsymbol{g}_{k-1,t}$ in the first inner product term on the right-hand side, and sum the inequalities over $t \in [T]$ with noticing $\boldsymbol{x}_{k-1} = \boldsymbol{x}_{k-1,1}$ and $\boldsymbol{x}_k = \boldsymbol{x}_{k-1,T+1}$, and obtain

$$T\big(f(\boldsymbol{x}_k) - f(\boldsymbol{z})\big) \leq \underbrace{\sum_{t=1}^{T}\langle \boldsymbol{g}_{k-1,t}, \boldsymbol{x}_k - \boldsymbol{x}_{k-1,t+1} \rangle - \frac{1}{2\eta_k}\sum_{t=1}^{T}\|\boldsymbol{x}_{k-1,t+1} - \boldsymbol{x}_{k-1,t}\|^2}_{\mathcal{T}_1}$$

$$+ \underbrace{\sum_{t=1}^{T}\langle \nabla f_t(\boldsymbol{x}_k) - \nabla f_t(\boldsymbol{x}_{k-1,t+1}), \boldsymbol{x}_k - \boldsymbol{x}_{k-1,t+1} \rangle}_{\mathcal{T}_2}$$

$$+ \underbrace{\sum_{t=1}^{T}\langle \nabla f_t(\boldsymbol{x}_{k-1,t+1}) - \boldsymbol{g}_{k-1,t}, \boldsymbol{x}_k - \boldsymbol{z} \rangle}_{\mathcal{T}_3} + \frac{1}{2\eta_k}\big(\|\boldsymbol{x}_{k-1} - \boldsymbol{z}\|^2 - \|\boldsymbol{x}_k - \boldsymbol{z}\|^2\big)$$

$$- \frac{1}{2L}\sum_{t=1}^{T}\Big(\|\nabla f_t(\boldsymbol{x}_k) - \nabla f_t(\boldsymbol{x}_{k-1,t+1})\|^2 + \|\nabla f_t(\boldsymbol{x}_{k-1,t+1}) - \nabla f_t(\boldsymbol{z})\|^2\Big).$$

For the term $\mathcal{T}_1$, we follow the argument from the proof of Theorem 1 to obtain

$$\mathcal{T}_1 = -\frac{1}{2\eta_k}\|\boldsymbol{x}_k - \boldsymbol{x}_{k-1}\|^2 \leq 0.$$

For the term $\mathcal{T}_2$, noticing that $\boldsymbol{x}_k - \boldsymbol{x}_{k-1,t+1} = -\eta_k \sum_{s=t+1}^{T} \boldsymbol{g}_{k-1,s}$ for $1 \leq t \leq T-1$ and $\boldsymbol{g}_{k-1,s} = \boldsymbol{g}_{k-1,s} - \nabla f_s(\boldsymbol{x}_{k-1,s+1}) + \nabla f_s(\boldsymbol{x}_{k-1,s+1}) - \nabla f_s(\boldsymbol{z}) + \nabla f_s(\boldsymbol{z}) - \nabla f_s(\boldsymbol{x}_*) + \nabla f_s(\boldsymbol{x}_*),$

we use Young's inequality with parameters $\alpha > 0$ and $\beta > 0$ to obtain

$$
\begin{aligned}
\mathcal{T}_2 &= \sum_{t=1}^{T-1} \Big\langle \nabla f_t(\boldsymbol{x}_k) - \nabla f_t(\boldsymbol{x}_{k-1,t+1}), -\eta_k \sum_{s=t+1}^{T} \boldsymbol{g}_{k-1,s} \Big\rangle \\
&\leq \frac{1}{2L}\Big(\frac{1}{2} + \frac{1}{\alpha} + \frac{1}{\beta}\Big) \sum_{t=1}^{T} \|\nabla f_t(\boldsymbol{x}_k) - \nabla f_t(\boldsymbol{x}_{k-1,t+1})\|^2 \\
&\quad + \frac{\alpha\eta_k^2 L}{2} \sum_{t=1}^{T-1} \Big\| \sum_{s=t+1}^{T} \big(\nabla f_s(\boldsymbol{z}) - \nabla f_s(\boldsymbol{x}_*)\big) \Big\|^2 + 2\eta_k^2 L \sum_{t=1}^{T-1} \Big\| \sum_{s=t+1}^{T} \nabla f_s(\boldsymbol{x}_*) \Big\|^2 \\
&\quad + \frac{\beta\eta_k^2 L}{2} \sum_{t=1}^{T-1} \Big\| \sum_{s=t+1}^{T} \big(\nabla f_s(\boldsymbol{x}_{k-1,s+1}) - \nabla f_s(\boldsymbol{z})\big) \Big\|^2 \\
&\quad + 2\eta_k^2 L \sum_{t=1}^{T-1} \Big\| \sum_{s=t+1}^{T} \big(\boldsymbol{g}_{k-1,s} - \nabla f_s(\boldsymbol{x}_{k-1,s+1})\big) \Big\|^2
\end{aligned}
$$

For the term $\mathcal{T}_3$, we use Cauchy-Schwarz inequality and Young's inequality to get

$$
\begin{aligned}
&\sum_{t=1}^{T} \langle \nabla f_t(\boldsymbol{x}_{k-1,t+1}) - \boldsymbol{g}_{k-1,t}, \boldsymbol{x}_k - \boldsymbol{z} \rangle \\
&\leq \frac{1}{2\sqrt{\eta_k}} \sum_{t=1}^{T} \|\nabla f_t(\boldsymbol{x}_{k-1,t+1}) - \boldsymbol{g}_{k-1,t}\| \|\boldsymbol{x}_k - \boldsymbol{z}\|^2 + \frac{\sqrt{\eta_k}}{2} \sum_{t=1}^{T} \|\nabla f_t(\boldsymbol{x}_{k-1,t+1}) - \boldsymbol{g}_{k-1,t}\|.
\end{aligned}
$$

Further using the fact that $\|\sum_{i=1}^{n} \boldsymbol{x}_i\|^2 \leq n \sum_{i=1}^{n} \|\boldsymbol{x}_i\|^2$ and combining the above bounds on $\mathcal{T}_1$, $\mathcal{T}_2$ and $\mathcal{T}_3$ with $\|\boldsymbol{g}_{k-1,t} - \nabla f_t(\boldsymbol{x}_{k-1,t+1})\| \leq \frac{\varepsilon_{k-1,t}}{\eta_k}$ for $t \in [T]$, we obtain

$$
\begin{aligned}
T\big(f(\boldsymbol{x}_k) - f(\boldsymbol{z})\big) &\leq \frac{1}{2L}\Big(\frac{1}{\alpha} + \frac{1}{\beta} - \frac{1}{2}\Big) \sum_{t=1}^{T} \|\nabla f_t(\boldsymbol{x}_k) - \nabla f_t(\boldsymbol{x}_{k-1,t})\|^2 \\
&\quad + \Big(\frac{\beta\eta_k^2 T^2 L}{2} - \frac{1}{2L}\Big) \sum_{t=1}^{T} \|\nabla f_t(\boldsymbol{x}_{k-1,t+1}) - \nabla f_t(\boldsymbol{z})\|^2 \\
&\quad + \frac{\alpha\eta_k^2 T^2 L}{2} \sum_{t=1}^{T} \|\nabla f_t(\boldsymbol{z}) - \nabla f_t(\boldsymbol{x}_*)\|^2 + 2\eta_k^2 L \sum_{t=1}^{T-1} \Big\| \sum_{s=t+1}^{T} \nabla f_s(\boldsymbol{x}_*) \Big\|^2 \\
&\quad + 2T^2 L \sum_{t=1}^{T} \varepsilon_{k-1,t}^2 + \frac{1}{2\eta_k^{3/2}} \sum_{t=1}^{T} \varepsilon_{k-1,t}\|\boldsymbol{x}_k - \boldsymbol{z}\|^2 + \frac{1}{2\sqrt{\eta_k}} \sum_{t=1}^{T} \varepsilon_{k-1,t} \\
&\quad + \frac{1}{2\eta_k}\big(\|\boldsymbol{x}_{k-1} - \boldsymbol{z}\|^2 - \|\boldsymbol{x}_k - \boldsymbol{z}\|^2\big).
\end{aligned}
$$

It remains to follow the proof of Lemma 1 and use $\eta_k \leq \frac{1}{\sqrt{\beta}TL} \leq \frac{1}{2TL}$ for $\beta \geq 4$ to obtain

$$
\begin{aligned}
T\big(f(\boldsymbol{x}_k) - f(\boldsymbol{z})\big) &\leq 2\eta_k^2 L \sum_{t=1}^{T-1} \Big\| \sum_{s=t+1}^{T} \nabla f_s(\boldsymbol{x}_*) \Big\|^2 + \frac{\alpha}{\beta} T\big(f(\boldsymbol{z}) - f(\boldsymbol{x}_*)\big) + \frac{T}{\eta_k} \sum_{t=1}^{T} \varepsilon_{k-1,t}^2 \\
&\quad + \frac{1}{2\eta_k}\|\boldsymbol{x}_{k-1} - \boldsymbol{z}\|^2 - \frac{1}{2\eta_k}\Big(1 - \frac{\sum_{t=1}^{T} \varepsilon_{k-1,t}}{\sqrt{\eta_k}}\Big)\|\boldsymbol{x}_k - \boldsymbol{z}\|^2 + \frac{\sum_{t=1}^{T} \varepsilon_{k-1,t}}{2\sqrt{\eta_k}},
\end{aligned}
$$

thus finishing the proof. $\qquad\square$

**Corollary 3** (Convex Smooth). *Under Assumptions 1 and 3, 4 and for parameters $\alpha, \beta$ such that $\frac{1}{\alpha} + \frac{1}{\beta} \leq \frac{1}{2}$, if the step size is fixed and satisfies $\eta_k \equiv \eta \leq \frac{1}{\sqrt{\beta}TL}$, the output $\boldsymbol{x}_K$ of Alg. 2 with inexact proximal point evaluations as in Eq. (8) with $\sum_{t=1}^{T} \varepsilon_{k-1,t} \leq \frac{\sqrt{\eta}}{1+(1+\alpha/\beta)(K-k+1)}$ satisfies*

$$
f(\boldsymbol{x}_K) - f(\boldsymbol{x}_*) \leq \frac{\mathrm{e}\|\boldsymbol{x}_0 - \boldsymbol{x}_*\|^2}{2\eta T K^{\frac{1}{1+\alpha/\beta}}} + 2\eta^2 T^2 \sigma_*^2 L(1+\beta/\alpha) K^{\frac{\alpha/\beta}{1+\alpha/\beta}} + \frac{\mathrm{e}}{2\eta T} \sum_{k=0}^{K-1} \sum_{t=1}^{T} \frac{2T\varepsilon_{k,t}^2 + \sqrt{\eta}\varepsilon_{k,t}}{(K-k)^{\frac{1}{1+\alpha/\beta}}}.
$$

Given $\epsilon > 0$, if $\sum_{t=1}^{T} \varepsilon_{k-1,t} \leq \sqrt{\eta} \min\{\varepsilon, \frac{1}{3(K-k+1)}\}$, there exists $\eta$ such that $f(\boldsymbol{x}_K) - f(\boldsymbol{x}_*) \leq \epsilon$ after $\widetilde{\mathcal{O}}\left(\frac{TL\|\boldsymbol{x}_0-\boldsymbol{x}_*\|^2}{\epsilon} + \frac{TL^{1/2}\sigma_*\|\boldsymbol{x}_0-\boldsymbol{x}_*\|^2}{\epsilon^{3/2}}\right)$ individual inexact proximal point evaluations.

*Proof.* Using Lemma 6 and following the proof of Theorem 1 with multiplying $\eta_k w_{k-1}$ on both sides, we have

$$T\eta_k w_{k-1}\big(f(\boldsymbol{x}_k) - f(\boldsymbol{z}_{k-1})\big)$$

$$\leq 2T^3 \eta_k^3 w_{k-1} L\sigma_*^2 + \frac{\alpha}{\beta} T\eta_k w_{k-1}\big(f(\boldsymbol{z}_{k-1}) - f(\boldsymbol{x}_*)\big) + Tw_{k-1}\sum_{t=1}^{T} \varepsilon_{k-1,t}^2 + \frac{w_{k-1}\sqrt{\eta_k}}{2}\sum_{t=1}^{T} \varepsilon_{k-1,t}$$

$$+ \frac{\lambda_{k-1}^2 w_{k-1}}{2}\|\boldsymbol{x}_{k-1} - \boldsymbol{z}_{k-2}\|^2 - \frac{w_{k-1}(1 - \sum_{t=1}^{T} \varepsilon_{k-1,t}/\sqrt{\eta_k})}{2}\|\boldsymbol{x}_k - \boldsymbol{z}_{k-1}\|^2.$$

Then we sum the above inequality over $k \in [K]$ and follow the proof of Theorem 1. To telescope the terms $\|\boldsymbol{x}_k - \boldsymbol{z}_{k-1}\|^2$, we need $\sum_{t=1}^{T} \varepsilon_{k-1,t}/\sqrt{\eta_k} \leq 1 - \lambda_k$ for $1 \leq k \leq K - 1$ such that

$$\lambda_k^2 w_k \leq \lambda_k w_k\Big(1 - \sum_{t=1}^{T} \varepsilon_{k-1,t}/\sqrt{\eta_k}\Big) \leq w_{k-1}\Big(1 - \sum_{t=1}^{T} \varepsilon_{k-1,t}/\sqrt{\eta_k}\Big).$$

In this case, we maintain the same requirements on $\{\lambda_k\}$ and $\{w_k\}$ to obtain the guarantee on the last iterate as in Theorem 1. In particular, we take the same choices with constant step sizes $\eta_k \equiv \eta$ such that $\lambda_k = \frac{w_{k-1}}{w_k} = \frac{(1+\frac{\alpha}{\beta})(K-k)}{1+(1+\frac{\alpha}{\beta})(K-k)}$ for $0 \leq k \leq K - 1$, so it suffices to let $\sum_{t=1}^{T} \varepsilon_{k-1,t} \leq \frac{\sqrt{\eta}}{1+(1+\frac{\alpha}{\beta})(K-k+1)}$ for $1 \leq k \leq K$. Following the proof of Theorem 1, we obtain

$$f(\boldsymbol{x}_K) - f(\boldsymbol{x}_*) \leq \frac{w_{-1}}{2\eta T}\|\boldsymbol{x}_0 - \boldsymbol{x}_*\|^2 + 2\eta^2 T^2 \sigma_*^2 L\sum_{k=1}^{K} w_{k-1}$$

$$+ \frac{1}{\eta}\sum_{k=1}^{K}\sum_{t=1}^{T} w_{k-1}\varepsilon_{k-1,t}^2 + \frac{1}{2\sqrt{\eta}T}\sum_{k=1}^{K}\sum_{t=1}^{T} w_{k-1}\varepsilon_{k-1,t}.$$

Plugging in the choice that $w_{k-1} \leq \frac{\mathrm{e}}{(K-k+1)^{\frac{1}{1+\alpha/\beta}}}$ for $1 \leq k \leq K - 1$ and $w_{K-1} = 1$, we then have

$$f(\boldsymbol{x}_K) - f(\boldsymbol{x}_*) \leq \frac{\mathrm{e}\|\boldsymbol{x}_0 - \boldsymbol{x}_*\|^2}{2\eta T K^{\frac{1}{1+\alpha/\beta}}} + 2\eta^2 T^2 \sigma_*^2 L(1+\beta/\alpha)K^{\frac{\alpha/\beta}{1+\alpha/\beta}} + \frac{\mathrm{e}}{2\eta T}\sum_{k=0}^{K-1}\sum_{t=1}^{T} \frac{2T\varepsilon_{k,t}^2 + \sqrt{\eta}\varepsilon_{k,t}}{(K-k)^{\frac{1}{1+\alpha/\beta}}}.$$

Hence, given $\varepsilon > 0$, to maintain the convergence rate with exact proximal point evaluations, it suffices to take $\sum_{t=1}^{T} \varepsilon_{k-1,t} \leq \sqrt{\eta}\min\{\frac{\varepsilon}{4\mathrm{e}^2(1+\log K)}, \frac{1}{1+(1+\frac{\alpha}{\beta})(K-k+1)}\}$ for $1 \leq k \leq K$. Indeed, we have

$$f(\boldsymbol{x}_K) - f(\boldsymbol{x}_*) \leq \frac{\mathrm{e}\|\boldsymbol{x}_0 - \boldsymbol{x}_*\|^2}{2\eta T K^{\frac{1}{1+\alpha/\beta}}} + 2\eta^2 T^2 \sigma_*^2 L(1+\beta/\alpha)K^{\frac{\alpha/\beta}{1+\alpha/\beta}} + \sum_{k=0}^{K} \frac{2\mathrm{e}\varepsilon}{(K-k)^{\frac{1}{1+\alpha/\beta}}}$$

$$\overset{(i)}{\leq} \frac{\mathrm{e}\|\boldsymbol{x}_0 - \boldsymbol{x}_*\|^2}{2\eta T K^{\frac{1}{1+\alpha/\beta}}} + 2\eta^2 T^2 \sigma_*^2 L(1+\beta/\alpha)K^{\frac{\alpha/\beta}{1+\alpha/\beta}} + 2\mathrm{e}\varepsilon(1+\beta/\alpha)K^{\frac{\alpha/\beta}{1+\alpha/\beta}}.$$

It remains to follow the proof of Theorem 1, and we choose $\sum_{t=1}^{T} \varepsilon_{k-1,t} = \sqrt{\eta}\min\{\varepsilon, \frac{1}{3(K-k+1)}\}$, assuming without loss of generality that $\varepsilon \leq \frac{1}{4\mathrm{e}^2(1+\log K)}$. □

We then come to prove the convergence with inexact proximal point evaluations for convex Lipschitz settings.

**Lemma 7.** *Under Assumptions [1] and [2], for any $\boldsymbol{z} \in \mathbb{R}^d$ that is fixed in the $k$-th cycle of Alg. [2], we have for $k \in [K]$*

$$
\begin{aligned}
T(f(\boldsymbol{x}_k) - f(\boldsymbol{z})) \leq{} & \frac{1}{2\eta_k}\Big(1 + \frac{1}{2\eta_k GT}\sum_{t=1}^{T}\varepsilon_{k-1,t}\Big)\|\boldsymbol{x}_{k-1} - \boldsymbol{z}\|^2 - \frac{1}{2\eta_k}\|\boldsymbol{x}_k - \boldsymbol{z}\|^2 \\
& + \frac{T(T-1)G^2\eta_k}{2} + \frac{1}{\eta_k}\Big(\sum_{t=1}^{T}\varepsilon_{k-1,t}\Big)^2 + 3GT\sum_{t=1}^{T}\varepsilon_{k-1,t}.
\end{aligned}
\tag{25}
$$

*Proof.* By Lipschitzness of each component function, we have for $t \in [T-1]$

$$
f_t(\boldsymbol{x}_k) - f_t(\boldsymbol{x}_{k-1,t+1}) \leq G\|\boldsymbol{x}_k - \boldsymbol{x}_{k-1,t+1}\| = G\eta_k\Big\|\sum_{s=t+1}^{T}\boldsymbol{g}_{k-1,s}\Big\|.
$$

Decomposing $\boldsymbol{g}_{k-1,s} = \boldsymbol{g}_{k-1,s} - \nabla M_{\eta_k f_s}(\boldsymbol{x}_{k-1,s}) + \nabla M_{\eta_k f_s}(\boldsymbol{x}_{k-1,s})$ and using triangle inequalities, we have

$$
\begin{aligned}
& f_t(\boldsymbol{x}_k) - f_t(\boldsymbol{x}_{k-1,t+1}) \\
& \leq \eta_k G\sum_{s=t+1}^{T}\Big(\|\boldsymbol{g}_{k-1,s} - \nabla M_{\eta_k f_s}(\boldsymbol{x}_{k-1,s})\| + \|\nabla M_{\eta_k f_s}(\boldsymbol{x}_{k-1,s})\|\Big) \\
& \overset{(i)}{\leq} \eta_k G\sum_{s=t+1}^{T}\Big(\frac{\varepsilon_{k-1,s}}{\eta_k} + G\Big) \leq (T-t)G^2\eta_k + G\sum_{s=t+1}^{T}\varepsilon_{k-1,s},
\end{aligned}
\tag{26}
$$

where we use Eq. (8) and the fact that $\nabla M_{\eta_k f_s}(\boldsymbol{x}_{k-1,s}) \in \partial f_t(\mathrm{prox}_{\eta_k f_t}(\boldsymbol{x}_{k-1,t}))$ for $(i)$. On the other hand, using convexity of $f_t$, we have for $t \in [T]$ that

$$
\begin{aligned}
f_t(\boldsymbol{z}) \geq{} & f_t(\boldsymbol{x}_{k-1,t+1}) + \langle\nabla M_{\eta_k f_t}(\boldsymbol{x}_{k-1,t}), \boldsymbol{z} - \boldsymbol{x}_{k-1,t+1}\rangle \\
={} & f_t(\boldsymbol{x}_{k-1,t+1}) + \langle\boldsymbol{g}_{k-1,t}, \boldsymbol{z} - \boldsymbol{x}_{k-1,t+1}\rangle + \langle\nabla M_{\eta_k f_t}(\boldsymbol{x}_{k-1,t}) - \boldsymbol{g}_{k-1,t}, \boldsymbol{z} - \boldsymbol{x}_{k-1,t+1}\rangle.
\end{aligned}
$$

Expanding the inner product in the above quantity and using Cauchy-Schwarz inequality with Eq. (8) leads to

$$
\begin{aligned}
& f_t(\boldsymbol{x}_{k-1,t+1}) - f_t(\boldsymbol{z}) \\
& \leq -\frac{1}{\eta_k}\langle\boldsymbol{x}_{k-1,t} - \boldsymbol{x}_{k-1,t+1}, \boldsymbol{z} - \boldsymbol{x}_{k-1,t+1}\rangle + \|\nabla M_{\eta_k f_t}(\boldsymbol{x}_{k-1,t}) + \boldsymbol{g}_{k-1,t}\|\|\boldsymbol{z} - \boldsymbol{x}_{k-1,t+1}\| \\
& \leq \frac{1}{2\eta_k}\big(\|\boldsymbol{x}_{k-1,t} - \boldsymbol{z}\|^2 - \|\boldsymbol{x}_{k-1,t+1} - \boldsymbol{z}\|^2\big) + \frac{\varepsilon_{k-1,t}}{\eta_k}\|\boldsymbol{x}_{k-1,t+1} - \boldsymbol{z}\|.
\end{aligned}
\tag{27}
$$

Using triangle inequalities and decomposing $\boldsymbol{x}_{k-1,t+1} - \boldsymbol{x}_{k-1} = \eta_k\sum_{s=1}^{t}\boldsymbol{g}_{k-1,s} - \nabla M_{\eta_k f_s}(\boldsymbol{x}_{k-1,s}) + \nabla M_{\eta_k f_s}(\boldsymbol{x}_{k-1,s})$, we bound the term $\mathcal{T} := \sum_{t=1}^{T}\varepsilon_{k-1,t}\|\boldsymbol{x}_{k-1,t+1} - \boldsymbol{z}\|$ in Eq. (27) as follows

$$
\begin{aligned}
\mathcal{T} \leq{} & \sum_{t=1}^{T}\varepsilon_{k-1,t}\big(\|\boldsymbol{x}_{k-1,t+1} - \boldsymbol{x}_{k-1}\| + \|\boldsymbol{x}_{k-1} - \boldsymbol{z}\|\big) \\
\leq{} & \sum_{t=1}^{T}\varepsilon_{k-1,t}\Big(\eta_k\sum_{s=1}^{t}\big(\|\boldsymbol{g}_{k-1,s} - \nabla M_{\eta_k f_s}(\boldsymbol{x}_{k-1,s})\| + \|\nabla M_{\eta_k f_s}(\boldsymbol{x}_{k-1,s})\|\big) + \|\boldsymbol{x}_{k-1} - \boldsymbol{z}\|\Big) \\
\leq{} & \sum_{t=1}^{T}\varepsilon_{k-1,t}\sum_{s=1}^{t}\varepsilon_{k-1,s} + \big(\eta_k GT + \|\boldsymbol{x}_{k-1} - \boldsymbol{z}\|\big)\sum_{t=1}^{T}\varepsilon_{k-1,t} \\
\overset{(i)}{\leq}{} & \Big(\sum_{t=1}^{T}\varepsilon_{k-1,t}\Big)^2 + 2\eta_k GT\sum_{t=1}^{T}\varepsilon_{k-1,t} + \frac{1}{4\eta_k GT}\sum_{t=1}^{T}\varepsilon_{k-1,t}\|\boldsymbol{x}_{k-1} - \boldsymbol{z}\|^2,
\end{aligned}
$$

where we use Young's inequality for $(i)$. Combining Eq. (26) and (27) with the above bound on $\mathcal{T}$ and noticing that $\boldsymbol{x}_{k-1,T+1} = \boldsymbol{x}_k$ and $\boldsymbol{x}_{k-1,1} = \boldsymbol{x}_{k-1}$, we sum the inequalities over $t \in [T]$ and obtain

$$T(f(\boldsymbol{x}_k) - f(\boldsymbol{z})) \le \frac{1}{2\eta_k}\Big(1 + \frac{1}{2\eta_k GT}\sum_{t=1}^{T}\varepsilon_{k-1,t}\Big)\|\boldsymbol{x}_{k-1} - \boldsymbol{z}\|^2 - \frac{1}{2\eta_k}\|\boldsymbol{x}_k - \boldsymbol{z}\|^2$$
$$+ \frac{T(T-1)G^2\eta_k}{2} + \frac{1}{\eta_k}\Big(\sum_{t=1}^{T}\varepsilon_{k-1,t}\Big)^2 + 3GT\sum_{t=1}^{T}\varepsilon_{k-1,t},$$

thus finishing the proof. $\qquad\square$

**Corollary 4** (Convex Lipschitz). *Under Assumptions 1 and 2, the output $\boldsymbol{x}_K$ of Alg. 2 with inexact proximal point evaluations as in Eq. (8) with $\sum_{t=1}^{T}\varepsilon_{k-1,t} \le \frac{\eta_k\eta_{k-1}GT}{\sum_{j=k}^{K}\eta_j}$ satisfies*

$$f(\boldsymbol{x}_K) - f(\boldsymbol{x}_*) \le \frac{\|\boldsymbol{x}_0 - \boldsymbol{x}_*\|^2}{2T\sum_{k=1}^{K}\eta_k} + \frac{G^2T}{2}\sum_{k=1}^{K}\frac{\eta_k^2}{\sum_{j=k}^{K}\eta_j} + \sum_{k=1}^{K}\sum_{t=1}^{T}\Big(\frac{\varepsilon_{k-1,t}^2}{2T\sum_{j=k}^{K}\eta_j} + \frac{3G\varepsilon_{k-1,t}\eta_k}{\sum_{j=k}^{K}\eta_j}\Big).$$

*Given $\epsilon > 0$, if $\sum_{t=1}^{T}\varepsilon_{k-1,t} \le \frac{2\eta GT}{K-k+1}$, there exists a constant step size $\eta$ such that $f(\boldsymbol{x}_K) - f(\boldsymbol{x}_*) \le \epsilon$ after $\widetilde{\mathcal{O}}\big(\frac{G^2T\|\boldsymbol{x}_0-\boldsymbol{x}_*\|^2}{\epsilon^2}\big)$ individual inexact proximal point evaluations.*

*Proof.* Using Lemma 7 with $\boldsymbol{z} = \boldsymbol{z}_{k-1}$ defined by Eq. (4) and multiplying $\eta_k w_{k-1}$ on both sides, we have

$$T\eta_k w_{k-1}(f(\boldsymbol{x}_k) - f(\boldsymbol{z}_{k-1}))$$
$$\le \frac{w_{k-1}\lambda_{k-1}^2(1 + \frac{1}{2\eta_k GT}\sum_{t=1}^{T}\varepsilon_{k-1,t})}{2}\|\boldsymbol{x}_{k-1} - \boldsymbol{z}_{k-2}\|^2 - \frac{w_{k-1}}{2}\|\boldsymbol{x}_k - \boldsymbol{z}_{k-1}\|^2$$
$$+ \frac{T(T-1)G^2\eta_k^2 w_{k-1}}{2} + \frac{w_{k-1}}{2}\Big(\sum_{t=1}^{T}\varepsilon_{k-1,t}\Big)^2 + 3GT\eta_k w_{k-1}\sum_{t=1}^{T}\varepsilon_{k-1,t}.$$

Then we sum the inequalities over $k \in [K]$ and follow the proof of Theorem 4. To telescope the terms $\|\boldsymbol{x}_k - \boldsymbol{z}_{k-1}\|^2$, we need $\lambda_{k-1} \le \frac{1}{1+\frac{1}{2\eta_k GT}\sum_{t=1}^{T}\varepsilon_{k-1,t}}$ for $1 \le k \le K-1$ such that

$$w_{k-1}\lambda_{k-1}^2(1 + \frac{1}{2\eta_k GT}\sum_{t=1}^{T}\varepsilon_{k-1,t}) \le w_{k-1}\lambda_{k-1} \le w_{k-2},$$

while we maintain other requirements on $\{\lambda_k\}$ and $\{w_k\}$ to obtain the last iterate convergence as in Theorem 4. In particular, we take the same choice that $w_k = \frac{\eta_K}{\sum_{j=k+1}^{K}\eta_j}$ and $\lambda_k = \frac{\sum_{j=k+1}^{K}\eta_j}{\sum_{j=k}^{K}\eta_j}$ for $0 \le k \le K-1$, so it suffices to let $\sum_{t=1}^{T}\varepsilon_{k-1,t} \le \frac{2\eta_k\eta_{k-1}GT}{\sum_{j=k}^{K}\eta_j}$. So we arrive at

$$f(\boldsymbol{x}_K) - f(\boldsymbol{x}_*) \le \frac{w_{-1}}{2T\eta_K}\|\boldsymbol{x}_0 - \boldsymbol{x}_*\|^2 + \frac{G^2T}{2\eta_K}\sum_{k=1}^{K}\eta_k^2 w_{k-1}$$
$$+ \frac{3G}{\eta_K}\sum_{k=1}^{K}w_{k-1}\eta_k\sum_{t=1}^{T}\varepsilon_{k-1,t} + \frac{1}{2T\eta_K}\sum_{k=1}^{K}\Big(\sum_{t=1}^{T}\varepsilon_{k-1,t}\Big)^2 w_{k-1}$$
$$= \frac{1}{2T\sum_{k=1}^{K}\eta_k}\|\boldsymbol{x}_0 - \boldsymbol{x}_*\|^2 + \frac{G^2T}{2}\sum_{k=1}^{K}\frac{\eta_k^2}{\sum_{j=k}^{K}\eta_j}$$
$$+ 3G\sum_{k=1}^{K}\sum_{t=1}^{T}\frac{\varepsilon_{k-1,t}\eta_k}{\sum_{j=k}^{K}\eta_j} + \frac{1}{2T}\sum_{k=1}^{K}\frac{(\sum_{t=1}^{T}\varepsilon_{k-1,t})^2}{\sum_{j=k}^{K}\eta_j}.$$

Hence, given $\varepsilon > 0$ and taking the constant step size $\eta_k \equiv \eta = \frac{\|\boldsymbol{x}_0 - \boldsymbol{x}_*\|}{GT\sqrt{K}}$ for simplicity, to maintain the convergence rate as in Theorem 4 with inexact proximal point evaluations, it suffices to let

$\sum_{t=1}^{T} \varepsilon_{k-1,t} \leq \frac{2\eta GT}{K-k+1}$. Indeed, we have

$$3G \sum_{k=1}^{K} \sum_{t=1}^{T} \frac{\varepsilon_{k-1,t}\eta_k}{\sum_{j=k}^{K}\eta_j} = 3G \sum_{k=1}^{K} \frac{\sum_{t=1}^{T}\varepsilon_{k-1,t}}{K-k+1} \leq \frac{5G\|\boldsymbol{x}_0 - \boldsymbol{x}_*\|}{\sqrt{K}},$$

and

$$\frac{1}{2T} \sum_{k=1}^{K} \frac{(\sum_{t=1}^{T}\varepsilon_{k-1,t})^2}{\sum_{j=k}^{K}\eta_j} \leq \frac{2G\|\boldsymbol{x}_0 - \boldsymbol{x}_*\|}{\sqrt{K}} \sum_{k=1}^{K} \frac{1}{(K-k+1)^3} \leq \frac{2.5G\|\boldsymbol{x}_0 - \boldsymbol{x}_*\|}{\sqrt{K}}.$$

It remains to follow the proof of Theorem 4, thus finishing the proof. $\qquad\square$

## D    CONCLUSION

This work provides the first oracle complexity guarantees for the last iterate of standard incremental (gradient and proximal) methods, motivated by catastrophic forgetting considerations in continual learning. The obtained complexity bounds nearly match the best known oracle complexity bounds that in the same settings were previously known only for the (uniformly) average iterate. Our results for the incremental proximal method further characterize the effect of regularization and its limitations in controlling catastrophic forgetting in continual learning applications. Our results complement prior theoretical findings in continual learning (CL) on linear models (Evron et al., 2022), where the authors enforce high task similarity in their setup and assume all tasks share a common set of loss minima. They further rely on implicit regularization to enforce closeness of models corresponding to subsequent tasks, arguing that this is sufficient for ensuring that forgetting will not be catastrophic. Our work explores the other side of the spectrum: if one does not assume tasks are "similar" (i.e., they do not necessarily have shared loss minima) but still enforces closeness of models using explicit regularization, it is impossible to prevent catastrophic forgetting entirely; however, one can ensure that forgetting is controlled using sufficient regularization. For future directions, it would be interesting to study what lies between the setups of prior work (Evron et al., 2022) and this work; in particular, to formally and quantitatively characterize the impact of (appropriately defined) task similarity on forgetting. Other possible future directions include 1) deriving last-iterate guarantees of incremental methods for nonconvex tasks; 2) obtaining convergence results matching ours in smooth convex settings while employing diminishing step sizes; and 3) deriving more fine-grained oracle complexity bounds by considering low-level stochastic oracles for each task and possible similarity between consecutive tasks.

