# OpenReview forum: "Last Iterate Convergence of Incremental Methods as a Model of Forgetting"
_ICLR.cc/2025/Conference — ICLR 2025 Poster_

### Official Review · Reviewer_kK5z · 2024-10-28

**Soundness:** 2
**Presentation:** 3
**Contribution:** 2
**Rating:** 6
**Confidence:** 3

**Summary:**

This paper addresses incremental gradient and proximal methods for optimization in convex settings, particularly focusing on the last-iterate convergence guarantees. Traditionally, convergence results for these methods have primarily focused on the average iterate. The authors present novel complexity bounds for the last iterate, which closely match those for the average iterate, extending these bounds to both incremental gradient descent (IGD) and incremental proximal methods (IPM) across various setups, including randomly permuted updates and weighted averaging schemes.

**Strengths:**

- The paper is clearly structured, with each theorem building on the previous results to form a coherent narrative.

- The paper provides novel non-asymptotic last-iterate convergence guarantees for incremental gradient and proximal methods without requiring strong convexity.

**Weaknesses:**

- While the paper discusses forgetting in the context of continual learning, the model of catastrophic forgetting is somewhat simplified (see questions).

- The paper presents theoretical contributions with lack of accompanying numerical experiments (only one with least square functions).

**Questions:**

- Why the last iterate convergence of the incremental proximal method is a good mathematical abstraction of forgetting in continual learning?

---

### Official Review · Reviewer_4LtA · 2024-11-02

**Soundness:** 4
**Presentation:** 2
**Contribution:** 3
**Rating:** 8
**Confidence:** 3

**Summary:**

The authors provide non-asymptotic finite-time optimality gap bounds for the final iterate of incremental gradient descent and incremental proximal method. These bounds match or improve on the sample complexity of the previous results obtained via a Polyak-Ruppert type iterate averaging.

**Strengths:**

- The authors address a fairly important setting in optimization and the last iterate optimality gap bounds are indeed new. In particular, the $\sqrt T$-improvement in sample complexity for IGD is a very nice result.

**Weaknesses:**

- I'm fairly sure this is standard and non-technical, but it would be great if bounds for decreasing step size without knowing the total number of rounds $K$ in advance were provided (at least in the appendix)

- The simulations are not entirely informative. Would it be possible to highlight the $\sqrt T$ improvement for IGD (I assume it might be possible since the authors' analysis indicates a different step size scheme compared to Mishchenko et al.)? Either way, I feel that removing the simulations and adding more summary/comparison/discussion for the theoretical results will strengthen the presentation of the paper.

**Questions:**

- There is a very minor notation inconsistency between Eq. (2) and Alg. 1 -- I think all the "k-1" subscripts in Alg. 1 can be "k" instead?

- In the two main theorems, I'd suggest stating the tuned gap bounds $ f(x_K) - f(x^*) \le $ [a decreasing function of $K$] as well.

- A clear comparison of results in a tabular form would contextualize the paper better.

- ''There were no guarantees for either the last iterate or even a weighted average of the iterates". Mishchenko et al.'s Section 3.1 provides a few bounds for the last iterate but these are in terms of the iterate error bounds instead of the optimality gap, right?

- The authors might want to comment after Cor. 2 on why the randomized version has a seemingly faster rate of convergence compared to its deterministic counterpart.

---

### Official Review · Reviewer_jyZz · 2024-11-03

**Soundness:** 3
**Presentation:** 2
**Contribution:** 3
**Rating:** 6
**Confidence:** 4

**Summary:**

This paper investigates the convergence properties of the last iterate in incremental methods. Specifically, it provides non-asymptotic convergence rates for the last iterate of incremental gradient descent and incremental proximal methods, with bounds that closely match those established for the average iterate. Additionally, the work characterizes the influence of regularization on the rate of forgetting within the convergence analysis of the incremental proximal method.

**Strengths:**

The paper presents novel results on the last-iterate convergence rates of incremental methods, which closely match the rates for average iterates. Additionally, it characterizes the effect of regularization on forgetting within the incremental proximal method.

**Weaknesses:**

The presentation of the paper requires improvement. Specifically, the explanation of the main proof techniques (Lemmas 1 and 2) is difficult to follow. Additionally, some of the parameter choices in the theorem statements are incomplete. Further details can be found in the Questions section.

**Questions:**

1. **Lemma 1**: What is the role of $\alpha$? Given that the choice of step size does not depend on $\alpha$, it seems feasible to set $\alpha$ as a constant throughout the paper.

2. **Lemma 2**: Is there a restriction on $\lambda_k$, similar to the restriction mentioned in line 238? After Lemma 2, it is stated that "the convergence rate for the last iterate is characterized by the growth rate of $\{w_k\}$”; however, this connection does not appear to be clearly reflected in the main results, such as Theorem 1. Moreover, lines 252-255 are hard to understand for me.

3. **Shuffled SGD**: Intuitively, random reshuffling and shuffling once generally do not yield similar convergence rates—random reshuffling reduces variance in gradient estimates each epoch, leading to faster and more stable convergence. In contrast, shuffling once fixes the sample order, lacking this ongoing variance reduction and potentially slowing convergence. Could the authors clarify why both random reshuffling and shuffling once are expected to yield similar rates?

4. **Theorems 2 & 3**: The proof for Theorem 2 is missing. Additionally, the current presentation of Theorems 2 and 3 suggests that a constant step size may not achieve the desired trade-off, implying a preference for a varying step size. Could similar results to those in Corollary 1 be established with a varying step size?

---

### Official Review · Reviewer_WMon · 2024-11-04

**Soundness:** 3
**Presentation:** 3
**Contribution:** 3
**Rating:** 8
**Confidence:** 3

**Summary:**

This paper studies the convergence of the last-iterate of incremental optimization methods (more specifically, incremental gradient descent and incremental proximal point methods) over convex functions, either Lipschitz smooth or Lipschitz continuous. Moreover, the paper proposes the setting of cyclic optimization of a finite-sum convex function with regularization as a theoretically-tractable model of Continual Learning with cyclic task replay to theoretically study the phenomenon of *forgetting* in CL.

On the optimization front, the authors show last-iterate convergence rates for both incremental methods they consider (only for smooth functions in the case of gradient descent, and for both non-smooth and smooth functions for the proximal method). The convergence rates they show nearly match the best-known rates for the convergence of the *average* iterate of these methods. When interpreting the proximal-point results as a model of CL, the authors

**Strengths:**

- Convergence rate of incremental/shuffled methods has been an active area of interest in the optimization for ML community, and last-iterate convergence is usually harder to pin-down if compared to the average iterate. Thus, the convergence results are certainly of interest to this community, and the proof techniques seem to involve some interesting techniques to get these rates;
- I am quite far from the Continual Learning community, but from my point-of-view it seems like an interesting theoretical proxy to study when and how catastrophic forgetting can happen, and the last-iterate convergence does yield some interesting insights;
- The presentation of this paper is very good. This is not a comment simply made to pad the "strengths" section: the authors do an amazing job in thoroughly discussing related work in a very instructive manner, bringing the reader up-to-speed on some of the most relevant results and settings quite fast, and most of the paper is quite well-organized and written.

**Weaknesses:**

I believe the weaknesses I found are somewhat minor, and some of them might be due simply to misunderstanding from my parts due to the fact that I did not have the time to really the paper very deeply (partially this is my own fault, since ICLR's reviewers this year had a light load).

- I think the main conclusion from the on the context of CL were too "isolated" from the rest of the literature. For example, it would have been interesting if the authors knew whether the behaviour of forgetting described in Theorem 2 (and by the discussion on regularization) was seen in empirical studies, or if there seems to be a disconnect (maybe due to the issue of task similarity that the authors mentioned). On a similar line, even though I haven't read the work of Evron et al. (2022), it seems like a discussion of the message in terms of CL should have been discussed and compared to the message from this paper. I understand that right before Sec 3.1 the authors do say that the work of Evron et al. is in a more restricted setting, while their results are more general but rely on regularization. Yet, I believe both try to study the same phenomenon (catastrophic forgetting in cyclic CL), and thus it seems that there could have been a discussion comparing these high-level conclusions between those two papers. This would be a necessarily speculative discussion, but it seems that there are more conclusions to be drawn from these results beyond the fact that the step-size cannot be too big nor too small to better control forgetting. If the authors believe I missed the main conclusion you wanted to draw from the CL connection, feel free to let me know;

- This is a minor point on presentation, but the proof sketch of Theorem 3 is quite crammed (most lines are stretched out because of the fraction being drawn inline) and it didn't seem to yield interesting insights about the proof strategy of Theorem 3. In fact, even though I didn't read in details, I decided to go to the appendix and skim the full proof of Theorem 3 instead of looking at the proof sketch in the main body, and I found it to be easier to read than the sketch, which is counter-intuitive. I am not sure what would be the ideal change in this case, and the authors should take the opinions of a time-constrained reviewer with a huge grain of salt.

---

**Summary of review**: I believe this paper is a solid contribution to the optimization literature in ML. Last-iterate convergence are at times technically challenging, and this paper presents interesting results on last-iterate convergence of incremental GD and proximal methods. Even though I don't feel confident to judge the relevance of the contributions on the CL part, the authors do a great job of describing the relevance of the model to CL and some of the conclusions we can draw from the theoretical results. Even though I felt the conclusions to CL could have been better developed, this is still a solid contribution and an easy accept on my opinion.

**Questions:**

- I am not sure if I got all the conclusions that the authors wanted to convey (or could convey) about the meaning of these results for CL. Are the main conclusions that catastrophic forgetting is inevitable in this setting (fixed step-size on cyclic IPM) and that there is a critical step-size to control forgetting, and anything smaller/bigger is worse? Does the work of Evron et al. (2022) reach different conclusions? (If I understood correctly, it likely does and it may have to do with similarity of the functions and shared minima).

- This might be way off since I only skimmed Evron et al. (2022), but they make a point of there being a distinction between the convergence to the offline optimum and forgetting, and I think I could not identify this distinction in this model. If I had more time I certainly could try to understand this by myself, but is there such a distinction on the model your paper studies?

- A small technical question, but I was curious whether the results of IPM on Lipschitz continuous functions could also be true for Incremental (sub)Gradient Descent. Is the main roadblock that the analysis for IPM follows from the connection to the gradient of the Moreau envelope?

---

### Meta-Review · Area_Chair_SNSg · 2024-12-24

**Metareview:**

The paper shows the convergence for last iterate for incremental gradient and proximal methods for smooth convex functions (for GD and proximal methods) and for convex functions (only for proximal methods). The oracle complexity for this matches (up to log factors) the oracle complexity of the average iterate for both classes of methods. The authors introduce interesting new techniques and the results are of interest to the community.

**Additional Comments On Reviewer Discussion:**

The reviewers were well engaged in discussions with authors.

---

### Decision · Program_Chairs · 2025-01-22

Accept (Poster)